# Live imaging molecular changes in junctional tension upon VE-cadherin in zebrafish

Anne Karine Lagendijk [1], Guillermo A. Gomez [2,3], Sungmin Baek [1], Daniel Hesselson [4], William E. Hughes[4], Scott Paterson[1], Daniel E. Conway [5], Heinz-Georg Belting [6], Markus Affolter [6], Kelly A. Smith [1], Martin A. Schwartz [7], Alpha S. Yap [2] & Benjamin M. Hogan [1]

Forces play diverse roles in vascular development, homeostasis and disease. VE-cadherin at endothelial cell-cell junctions links the contractile acto-myosin cytoskeletons of adjacent cells, serving as a tension-transducer. To explore tensile changes across VE-cadherin in live zebrafish, we tailored an optical biosensor approach, originally established in vitro. We validate localization and function of a VE-cadherin tension sensor (TS) in vivo. Changes in tension across VE-cadherin observed using ratio-metric or lifetime FRET measurements reflect acto-myosin contractility within endothelial cells. Furthermore, we apply the TS to reveal biologically relevant changes in VE-cadherin tension that occur as the dorsal aorta matures and upon genetic and chemical perturbations during embryonic development.

[1] Institute for Molecular Bioscience, Genomics of Development and Disease division, The University of Queensland, 306 Carmody Road, St Lucia, 4072 QLD, Australia. [2] Institute for Molecular Bioscience, Cell Biology and Molecular Medicine division, The University of Queensland, 306 Carmody Road, St Lucia, 4072 QLD, Australia. [3] Centre for Cancer Biology, SA Pathology and the University of South Australia, Frome Road, Adelaide, 5000 SA, Australia. [4] Garvan Institute of Medical Research, 384 Victoria Street, Darlinghurst, Sydney, 2010 NSW, Australia. [5] Department of Biomedical Engineering, Virginia Commonwealth University, Richmond, VA 23284, USA. [6] Biozentrum der Universität Basel, Klingelbergstrasse 70, 4056 Basel, Switzerland. [7] Yale Cardiovascular Research Center and Department of Internal Medicine, Cardiovascular Medicine, Yale University School of Medicine, New Haven, CT 06510, USA. Alpha S. Yap and Benjamin M. Hogan contributed equally to this work. Correspondence and requests for materials should be addressed to A.K.L. (email: a.lagendijk@imb.uq.edu.au)

The involvement of mechanical forces and mechanotransduction in building and maintaining the vasculature is well appreciated. Changes in adhesive forces, fluid shear, intraluminal pressure and matrix stiffness have been shown to have profound impacts on vascular development and homeostasis[1–4]. In the vasculature, vascular endothelial (VE)-cadherin not only provides adhesive strength but VE-cadherin itself is under tension[5] and takes part in the endothelial mechanosensation complex, together with PECAM-1 and VEGFR2/3[6, 7]. This complex is required for sensing fluid shear stress in endothelial cellular monolayers and triggering the stereotypical changes in cellular morphology and vessel structure that are seen upon increased shear[6, 7]. Shear stress increases tension across PECAM-1 through enhanced association with Vimentin[5]. By contrast, increased shear in cultured cells decreases tension through VE-cadherin. This occurs whilst endothelial cells (ECs) respond to shear by aligning with the direction of flow in culture conditions[5]. Much remains to be understood about how mechanical forces in

**Fig. 1** VE-cadherin-TS is localized to cell-cell junctions and reports a differential FRET signature. **a** Schematic representation of the tension sensor (TS) module. The Donor fluorescent protein, Teal, is separated from the acceptor, Venus, by a stretchable linker peptide (top). When the module is under tension (bottom), energy transfer from Teal to Venus will decrease. **b** Schematic of zebrafish *VE-cadherin-TS* cDNA recombined into the *ve-cadherin* BAC clone. **c** Venus expression from *Tg(ve-cad:ve-cadTS)* throughout the blood vasculature at 2 dpf. Scale bar = 150 μm. **d** Maximum projection of the dorsal aorta (DA) at 3 dpf showing expression of Teal (top) and Venus (bottom) localized at cell–cell junctions. Scale bar = 10 μm. **e** Co-expression (Merge, bottom) of Venus (yellow, top) from *Tg(ve-cad:ve-cadTS)* and *Tg(fli1ep:lifeact-mCherry)*, labeling F-actin (purple, middle), in endothelial junctions at 3 dpf. Scale bar = 5 μm. **f–g** Fluorescence intensity of Venus (**f**) and *Tg(fli1ep:lifeact-mCherry)*, labeling F-actin (**g**), along a single 3 μm line-region in a single Z-section (shown in **e**). **h** RAW FRET signal at endothelial junctions at 2 dpf. Scale bar = 5 μm. **i** Ratio-metric FRET signal at 2 dpf in endothelial junctions (shown in **h**). Scale bar = 5 μm. **j** Variable ratio-metric FRET values along a single 10 μm line-region in a single Z-section in a proportion of the junction (boxed in **i**)

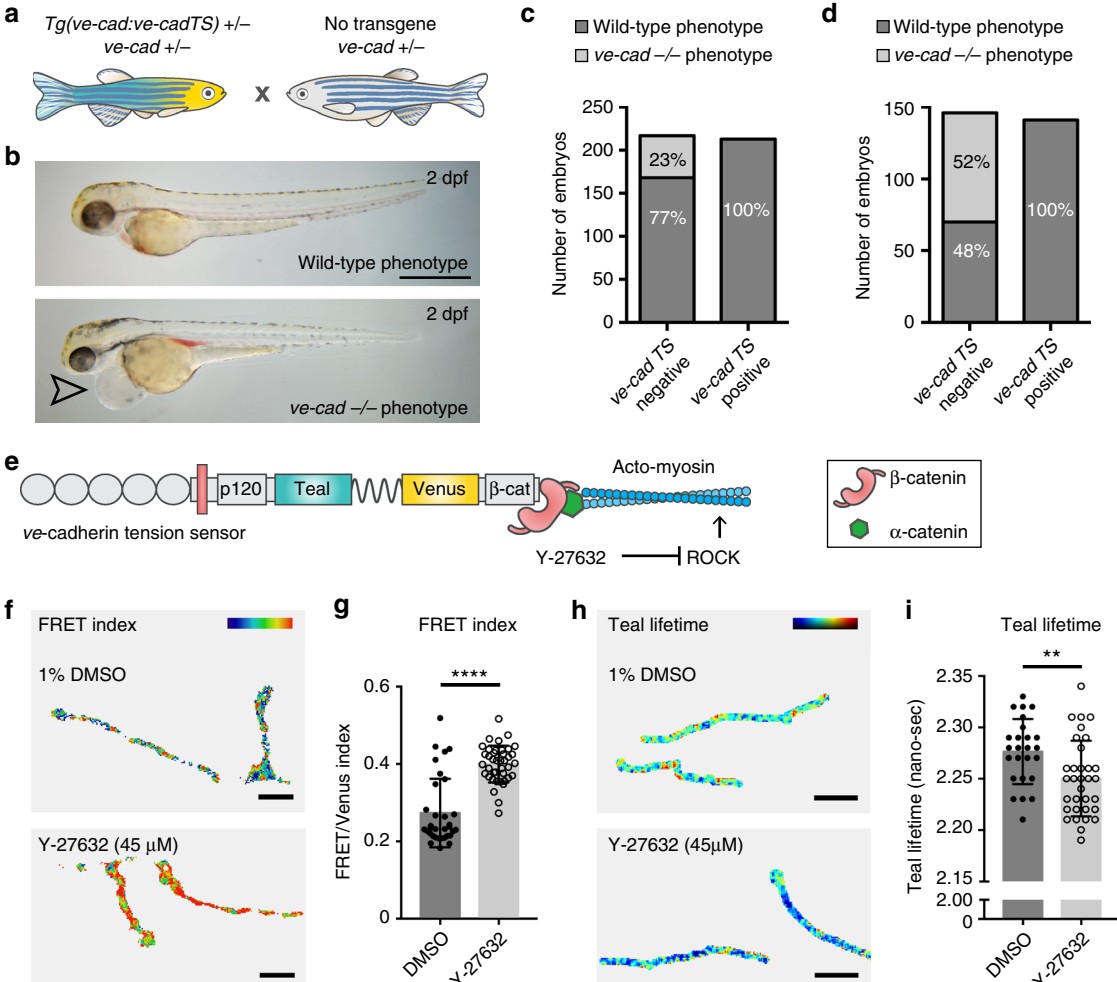

**Fig. 2** VE-cadherin-TS is functional and under acto-myosin controlled tension. **a** Schematic representation of genetic cross to test functionality of VE-cadherin-TS. **b** Wild-type (top) and *ve-cadherin* mutant (bottom) phenotype at 2 dpf (arrowhead indicates cardiac oedema in mutant). Scale bar = 1 mm. **c** Phenotypic scoring all embryos (n = 430) collected from cross in a, showing 23% phenotypic mutants (n = 49/217) in TS-negative population (n = 217) and 0% (n = 0/213) in TS-positive siblings (n = 213). **d** Phenotypic scoring of n = 286 embryos from a cross between a Tg(ve-cad:ve-cadTS)$^{+/-}$, cdh5$^{ubs8-/-}$ mutant and a non-transgenic cdh5$^{ubs8+/-}$ animal. In the TS negative population 52% (n = 76/146) displayed the mutant phenotype whilst there were no phenotypic mutants in the TS positive clutch (n = 141). **e** Schematic representation of Y-27632 (ROCK inhibitor) acto-myosin inhibition. **f** Heatmap image of ratio-metric FRET values in junctions of 1%DMSO and Y-27632 (45 μM/1%DMSO) treated embryos. Colors range from blue ( = low FRET index/high tension) to red ( = high FRET index/low tension). Scale bar = 5 μm. **g** Quantification of ratio-metric FRET values in junctions of 1 % DMSO (n = 4) versus Y-27632 (45 μM/1%DMSO) (n = 5) treated embryos. Each data point represents a junctional region of interest (ROI), n = 33 junctions selected from DMSO control embryos and n = 40 selected in Y-27632 treated embryos. Treatment for 5 h prior to imaging. Error bars represent mean ± s.d.; ****p < 0.0001 from unpaired two-sided Mann–Whitney test. **h** Heatmap image of Teal lifetime values in junctions of 1%DMSO (top) and Y-27632 (bottom, 45 μM/1%DMSO)- treated embryos. Colors range from blue ( = low lifetime/low tension) to red ( = high lifetime/high tension). Scale bar = 5μm. **i** Quantification of Teal lifetime values (nano-sec) in junctions of 1 % DMSO (n = 4) versus Y-27632 (45 μM/1%DMSO) (n = 4) treated embryos. Each data point represents a single junctional ROI, n = 25 junctions segmented from DMSO control embryos and n = 33 from Y-27632 treated embryos. Treatment for 5 h prior to imaging. Error bars represent mean ± s.d.; **p = 0.0060 from unpaired two-sided t test

the vasculature are sensed in vivo, transduced and how forces control the morphogenesis of complex vascular networks.

Currently, observing and quantifying the forces that act in cells and tissues, including the vasculature, is a significant technical challenge. Studies in cultured cells have pioneered our understanding of how physical forces act upon cells and control biological processes. However, approaches used to measure force in vitro such as optical and magnetic tweezers, micro-pipet aspiration or atomic force microscopy are disruptive and are not broadly applicable in vivo. Tension sensor (TS) technology has introduced an optical imaging-based read-out for tension across individual molecules in cells[5, 8–10]. Such imaging-based quantification of intra-molecular tension through key adhesive

molecules holds great promise for direct observation of forces that control morphogenesis. Recent applications of functionally validated TS-proteins for β-spectrin in *Caenorhabditis elegans*[11] and E-cadherin in *Drosophila*[12] have confirmed the potential of measuring intra-molecular tension changes in invertebrate models. Nevertheless, how broadly applicable such approaches will prove in vertebrate animal models remains to be determined.

Here we successfully applied the TS approach in the tubular vasculature of a living vertebrate by generating a zebrafish VE-cadherin-TS strain. Our findings identify changes in VE-cadherin tension that occur during arterial maturation, highlight key controls for the validation of TS readouts in vivo and indicate the utility of the approach for phenotyping in genetic and pharmaco-

genetic perturbations. We suggest that TS readouts hold great promise for broad application in in vivo mechanobiology studies but that careful technical controls and validation are required.

## Results

**A zebrafish VE-cadherin TS transgenic strain.** A TS module that contains two fluorescent proteins (Teal and Venus), separated by an elastic entropic linker peptide, forms the core component of the approach[8, 13, 14]. Stretching or relaxation of the linker can be quantified by Förster resonance energy transfer (FRET)[8, 13] (Fig. 1a). We cloned this TS module into the zebrafish VE-cadherin cDNA, between the p120-catenin and β-catenin binding sites of the translated protein, as previously described for mouse VE-cadherin in vitro[5] (Fig. 1b, Supplementary Fig. 1a). To express the module at levels resembling endogenous VE-cadherin, we recombined it into a *ve-cadherin* BAC[15] (Fig. 1b). The transgenic line *TgBAC(ve-cad:ve-cadTS)*[uq11bh] (hereafter *Tg(ve-cad:ve-cadTS)*) expressed VE-cadherin-TS at cell–cell junctions of the vasculature and endocardium (Fig. 1c, d and Supplementary Fig. 1b), allowing visualization of junctional VE-cadherin at unprecedented resolution in live zebrafish. VE-cadherin-TS expression co-localized with junctional filamentous (F)-actin, visualized by *Tg(fli1ep:lifeact-mCherry)*[uq12bh] confirming correct localization of the protein (Fig. 1e–g). We could also detect a junctional FRET signal (Fig. 1h). To correct the FRET signal for fluorescence bleed-through (BT), we generated two additional transgenic lines, *Tg(10xUAS:Teal)*[uq13bh] and *Tg(10xUAS:Venus)*[uq8bh] (Supplementary Fig. 1c). We used the endothelial *TgBAC (ve-cad:GALFF)*[15] driver to express the fluorophores individually and measure BT values for each (Supplementary Fig. 1c). This allowed us to calculate ratio-metric energy transfer through VE-cadherin as the ratio between BT-corrected FRET signal and Venus fluorescence, which at the same time normalizes for the amount of VE-cadherin-TS protein (Fig. 1i,j and Supplementary Fig. 1d). Importantly, correlation analysis comparing FRET index values to Venus fluorescence (i.e. protein expression) in location-matched pixels within single junctions (Supplementary Fig. 1e–h') or in a grouped analysis of all pixels from individual junctions (Supplementary Fig. 1i) revealed that there is no positive correlation between FRET values and TS protein concentration (Supplementary Fig. 1e–i), indicating that there is no significant contribution of inter-molecular energy transfer[16–20]. These results suggest the utility of this transgenic line to detect local differences in VE-cadherin tension along individual cell–cell junctions in a live vertebrate.

**VE-cadherin mutants are rescued by VE-cadherin-TS.** To test functionality of the VE-cadherin-TS protein, we crossed *Tg(ve-cad:ve-cadTS)* onto a *ve-cadherin* mutant background, *cdh5(ve-cad)*[ubs8][21]. Loss of VE-cadherin function results in cardiovascular failure[21, 22]. We crossed heterozygous carriers for *Tg(ve-cad:ve-cadTS)*[+/−] and *cdh5*[ubs8+/−] to non-transgenic heterozygous carriers for *cdh5*[ubs8+/−] (Fig. 2a). We separated TS-positive and TS-negative progeny at 1 dpf, prior to the onset of phenotypic abnormalities. At 2 days post fertilisation (dpf), phenotypic mutants can be distinguished by the appearance of cardiovascular defects (Fig. 2b). Scoring revealed no phenotypic mutants in *Tg(ve-cad:ve-cadTS)*-positive progeny (Fig. 2c). The *Tg(ve-cad:ve-cadTS)*-negative siblings contained mutant phenotypes at a mendelian ratio (Fig. 2c). Finally, *Tg(ve-cad:ve-cadTS)*[+/−], *cdh5*[ubs8−/−] mutant fish were adult viable (Supplementary Fig. 2a) and crossing these to non-transgenic *cdh5*[ubs8+/−] carriers resulted in 50% phenotypic mutants in *Tg(ve-cad:ve-cadTS)*-negative progeny and no phenotypes in *Tg(ve-cad:ve-cadTS)*-positive progeny (Fig. 2d). We next examined the amount of VE-cadherin

protein (by quantification of Venus fluorescence intensity) at junctions in heterozygous and mutant animals (all *Tg(ve-cad:ve-cadTS)*[+/−]), and we found that the gene dosage of the wild-type VE-cadherin allele influences the amount of transgene derived VE-cadherin at the junctions. This was observed as higher Venus fluorescent intensities, at junctions in homozygous mutants (Supplementary Fig. 2b) This observation indicates that ECs actively regulate functional VE-cadherin levels at their adherens junctions and suggests that forced overexpression of TS constructs is unlikely to generate data reflecting the normal physiologically regulated conditions. Importantly, these results show that VE-cadherin-TS protein can compensate for loss of endogenous VE-cadherin, effectively reconstituting an endogenous protein with a synthetic TS version.

**FRET in VE-cadherin-TS relies on acto-myosin contractility.** To determine if VE-cadherin-TS reports changes in acto-myosin activity, the principal source of contractile force within cells, we generated a negative control sensor lacking the β–catenin-binding region of the cadherin cytoplasmic tail. This is predicted to be incapable of sensing acto-myosin generated force (referred to as tail less (TL)) and was previously used to verify a VE-cadherin-TS in vitro[5] (Supplementary Fig. 2c). We injected *ve-cadherin-TS* and *ve-cadherin-TL* DNA and quantified FRET index values, observing significantly higher ratio-metric FRET (decreased tension) in VE-cadherin-TL compared with TS expressing junctions (Supplementary Fig. 2d–e). However, in performing these control experiments, we observed that forced expression of BAC DNA frequently resulted in protein aggregation and mislocalisation for the TL but not the TS construct, likely due to the incapacity of this construct to bind β-catenin (Supplementary Fig. 2f–g). Furthermore, the TS but not the TL construct generated stable germline transmitted embryos despite large numbers of injected animals being tested for the later.

As a further control for the acto-myosin contribution to TS changes, we analysed embryos expressing the stable TS transgene upon chemical inhibition (by Y-27632) of Rho-associated kinase (ROCK), which is essential for acto-myosin contraction (Fig. 2e and Supplementary Fig. 2h). We detected a reproducible significant increase in ratio-metric FRET in junctions of Y-27632-treated embryos (Fig. 2f, g) as also observed in vitro[5]. Treatment with Y-27632 was carefully titrated to avoid induction of developmental defects. Therefore, while it acts to reduce acto-myosin contractility it may not be a fully inhibitory dosage. We further identified a significant increase in ratio-metric FRET upon chemical inhibition of the GTPase Cdc42, an additional regulator of acto-myosin assembly and junctional integrity[23] (Supplementary Fig. 2i–k). To control for the possibility that the changes observed were due to the ratio-metric FRET imaging approach being used, we measured tension changes upon Y-27632 treatment using donor fluorescence lifetime measurements (FLIM), a quantitative imaging technique that is less sensitive to imaging or processing artifacts than the ratio-metric approach[17, 24, 25]. In accordance with the ratio-metric analysis, VE-cadherin-TS in Y-27632-treated embryos was under significantly less tension (lower Teal lifetime) compared to DMSO-treated controls (Fig. 2h, i). Altogether these data demonstrate that differences in energy transfer in this transgenic strain reflect acto-myosin supported changes in tension through VE-cadherin in vivo. Furthermore, these observations establish Y-27632 treatment as a suitable and reliable negative control for biological experiments that utilize the VE-cadherin-TS strain.

**VE-cadherin tension decreases in maturing DA junctions.** To test the use of this strain to report biologically informative

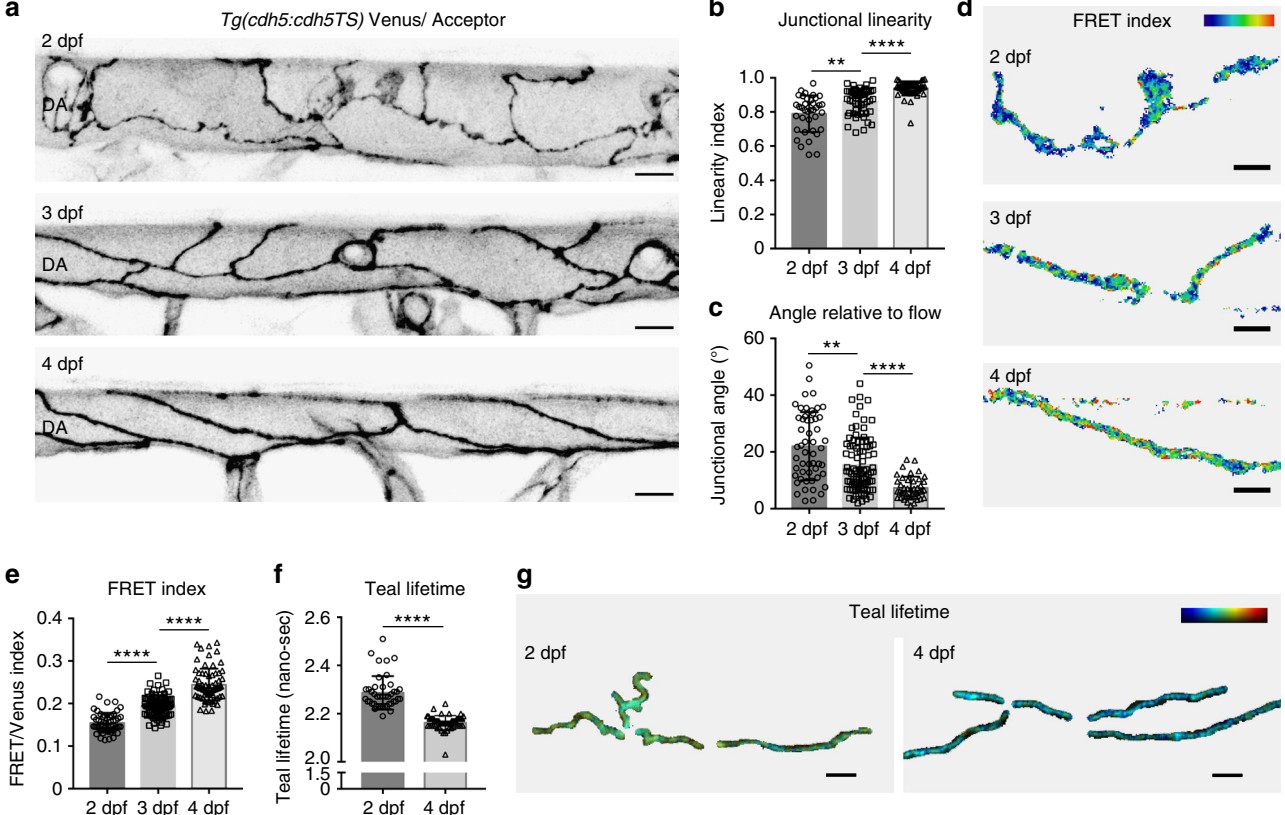

**Fig. 3** Cellular morphology and VE-cadherin tensile changes occur during artery maturation. **a** Junctional morphology of ECs in the dorsal aorta (DA) (Venus, grey) at 2 dpf (top), 3 dpf (middle) and 4 dpf (bottom). Scale bar = 10 μm. **b** Junctional linearity index over time at 2 dpf ($n = 37$ junctions from $n = 4$ embryos), 3 dpf ($n = 37$ junctions from $n = 5$ embryos) and 4 dpf ($n = 51$ junctions from $n = 4$). Error bars represent mean ± s.d.; 2–3 dpf **$p = 0.0010$; 3–4 dpf ****$p < 0.0001$, from unpaired two-sided Mann–Whitney test. **c** Junctional angle relative to the direction of blood flow over time at 2 dpf ($n = 53$ junctions from $n = 4$ embryos), 3 dpf ($n = 89$ junctions from $n = 5$ embryos) and 4 dpf ($n = 42$ junctions from $n = 4$). Error bars represent mean ± s.d.; 2–3 dpf **$p = 0.0010$; 3dpf– 4dpf ****$p < 0.0001$ from unpaired two-sided Mann–Whitney test. **d** Heatmap image of ratio-metric FRET values in junctions at 2 dpf (top), 3 dpf (middle) and 4 dpf (bottom). Colors range from blue ( = low FRET index/high tension) to red ( = high FRET index/low tension). Scale bar = 5 μm. **e** Quantification of ratio-metric FRET values in junctions at 2 dpf ($n = 54$ junctional ROIs from $n = 4$ embryos), 3 dpf ($n = 79$ junctional ROIs from $n = 5$ embryos) and 4 dpf ($n = 71$ junctional ROIs from $n = 4$ embryos). Error bars represent mean ± s.d.; 2–3 dpf ****$p < 0.0001$, from unpaired two-sided t test; 3–4 dpf ****$p < 0.0001$ from unpaired two-sided Mann–Whitney test. **f** Quantification of Teal lifetime values (nano-sec) comparing $n = 43$ junctional ROIs segmented from 2 dpf embryos ($n = 8$) and $n = 54$ junctional ROIs from 4 dpf embryos ($n = 10$). Error bars represent mean ± s.d.; ****$p < 0.0001$ from unpaired two-sided Mann–Whitney test. **g** Heatmap image of Teal lifetime values in junctions of 2 dpf (left) versus 4 dpf (right) embryos. Colors range from blue ( = low lifetime/low tension) to red ( = high lifetime/high tension). Scale bar = 5 μm

changes in vessels, we examined mechanical changes at EC junctions during artery maturation. From 2, the main axial artery, the dorsal aorta (DA), undergoes observable morphological changes, most prominently a decrease in lumen diameter[26, 27]. By visualizing VE-cadherin-TS localization over time, we observed striking morphological changes of the EC junctions from 2 to 4 dpf (Fig. 3a). At 2 dpf, junctions were irregular and immature compared with junctions at 4 dpf, and with development the junctions became progressive more linear and aligned with the direction of blood flow (Fig. 3b, c). These cellular rearrangements have been suggested to be induced by changes in flow and are necessary for the reduction of vessel diameter seen during maturation[27]. Over the period from 2 to 4 dpf, VE-cadherin-TS ratio-metric FRET measurements reported a robust and significant decrease in tension across VE-cadherin (Fig. 3d, e). We further confirmed this observation using FLIM measurements, which revealed reduced intramolecular tension in 4 dpf linear junctions compared with immature 2 dpf junctions (Fig. 3f, g). Of note, the differences in FRET efficiency measured by FLIM were small. The efficiency change for Y-27632 compared with DMSO was 1% and comparing 2 dpf to 4 dpf junctions was 4% (see "Methods" section).

Interestingly, cultured ECs align with the direction of flow when under shear[28] and this was shown to coincide with decreased tension through VE-cadherin[5], measured by FLIM as a 4% increase in FRET efficiency using a VE-cadherin-TS. This data is in line with the efficiency changes measured here in aligned, linear junctions at 4 dpf compared to immature junctions at 2 dpf. In addition, previously published comparison of a negative control construct with the VE-cadherin-TS revealed only a 7% change in FRET efficiency[5] measured by FLIM. Together, these observations suggest that the dynamic range for this VE-cadherin-TS is limited. Nevertheless, these data combined demonstrate that we can measure meaningful changes in tension through VE-cadherin in vivo. Our results indicate that there is a correlation between the junctional rearrangements that occur to allow DA constriction[27] and mechanical changes though VE-cadherin at the junctions.

### Vegf regulates morphological and mechanical DA maturation.
Vegfr2 activity plays diverse roles in the development and function of the vasculature that include the regulation of arterial angiogenesis, vascular permeability and junctional state[29].

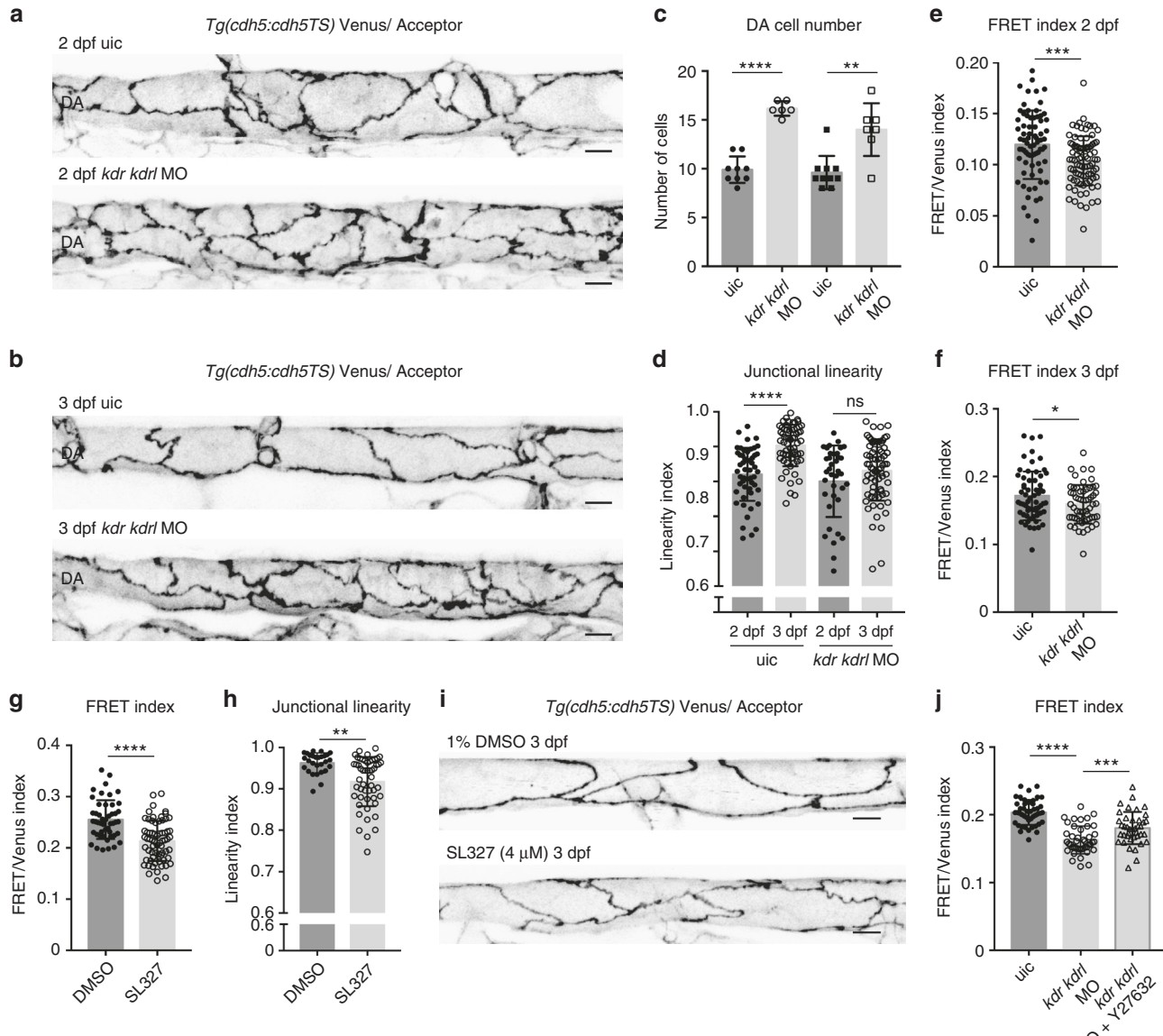

**Fig. 4** Inhibiting Vegf signaling impairs linearisation and VE-cadherin tension changes during arterial maturation. (**a**, **b**) Junctional morphology of ECs in the DA (Venus, grey) in uic (top) and *kdr/kdrl* morphants (bottom) at 2 dpf (**a**) and 3 dpf (**b**). Scale bar = 10 µm. **c** Number of ECs quantified at 2 dpf (uic *n* = 9, *kdr/kdrl* MO *n* = 6 embryos) and 3 dpf (uic *n* = 10, *kdr/kdrl* MO *n* = 7 embryos). Error bars represent mean ± s.d.; 2 dpf uic–2 dpf *kdr/kdrl* MO ****$p$ < 0.0001 from unpaired two-sided *t* test, 3 dpf uic–3 dpf *kdr/kdrl* MO **$p$ = 0.0068 from unpaired two-sided Mann–Whitney test. **d** Junctional linearity index of junctions in uic and *kdr/kdrl* morphants at 2 dpf and 3 dpf. Error bars represent mean ± s.d.; 2dpf uic–3dpf uic ****$p$ < 0.0001; 2 dpf *kdr/kdrl* MO–3dpf *kdr/kdrl* MO, $p$ = 0.0862 from unpaired two-sided Mann–Whitney test. **e**, **f** Ratio-metric FRET values of uic and *kdr/kdrl* morphants at 2 dpf **e** (uic *n* = 74 junctional ROIs from *n* = 9 embryos, *kdr/kdrl* MO *n* = 86 junctional ROIs from *n* = 6 embryos) and 3 dpf **f** (uic *n* = 62 junctional ROIs from *n* = 10 embryos, *kdr/kdrl* MO *n* = 58 junctional ROIs from *n* = 7 embryos). Error bars represent mean ± s.d.; ***$p$ = 0.0007; *$p$ = 0.0391 from unpaired two-sided *t* test. **g** Ratio-metric FRET values from 1% DMSO controls (*n* = 48 junctional ROIs from *n* = 6 embryos) versus SL327 (4 µM/1% DMSO)-treated embryos (*n* = 66 junctional ROIs from *n* = 10 embryos). Treatment from 20 hpf to 3 dpf. Error bars represent mean ± s.d.; ****$p$ < 0.0001 from unpaired two-sided *t* test. **h** Junctional linearity index in 1% DMSO treated controls (*n* = 32 junctional ROIs from *n* = 6 embryos) versus SL327 (4 µM/1% DMSO)-treated embryos (*n* = 56 junctional ROIs from *n* = 10 embryos). Error bars represent mean ± s.d.; **$p$ = 0.0045 from unpaired two-sided Mann–Whitney test. **i** Junctional morphology of ECs in the DA (Venus, grey) at 3 dpf in 1% DMSO and SL327 (4 µM/1% DMSO) treated embryos at 3 dpf. Scale bar = 10 µm. **j** Ratio-metric FRET values from uic (*n* = 45 junctional ROIs from *n* = 8 embryos) versus *kdr/kdrl* morphants incubated in 1% DMSO as a control (*n* = 43 junctional ROIs from *n* = 9 embryos) and *kdr/kdrl* Y-27632-treated morphants (50 µM/1% DMSO, *n* = 40 junctional ROIs from *n* = 9 embryos) at 2 dpf. Error bars represent mean ± s.d.; uic–*kdr/kdrl*MO + DMSO ****$p$ < 0.0001; *kdr/kdrl*MO + DMSO – *kdr/kdrl*MO + Y-27632 ***$p$ = 0.0005 from unpaired two-sided *t* test

Furthermore, together with VE-cadherin and PECAM-1, Vegfr2 takes part in the mechanosensory complex and the EC response to flow[6]. To investigate if interfering with Vegfr2-signaling would influence maturation of the DA, we knocked down both *kdr* and *kdrl* (zebrafish Vegfr2 homologues) using morpholinos (MOs) and explored the impact on the mechanical state of junctions in the DA over time. Sub-optimal dosages were used to inhibit arterial sprouting without affecting blood flow (Supplementary Fig. 3a and Supplementary Movie 1). These MOs phenocopy demonstrated mutant phenotypes for *kdr/kdrl*

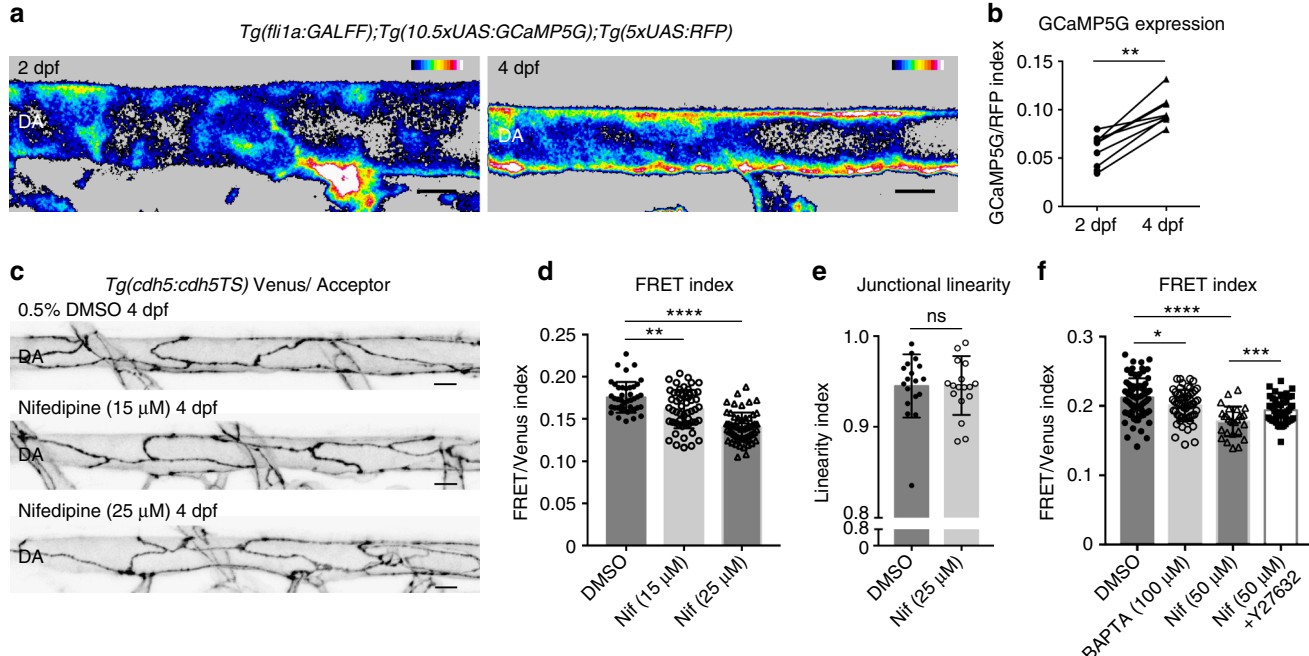

**Fig. 5** Calcium signaling maintains lower VE-cadherin tension during artery maturation. **a** Heatmap image showing average intensity projection of *fli1a*-driven GCaMP5G expression (indicating calcium) in the DA at 2 dpf (left) and 4 dpf (right). Colors range from black (= low calcium) to white (= high calcium). Scale bar = 10 μm. **b** Increase in fluorescence intensity index (= *fli1a* driven GCaMP5G/ *fli1a*-driven RFP) in the dorsal wall of the DA of $n = 9$ embryos imaged at 2 and 4 dpf. **p = 0.0012, from paired $t$ test. **c** Junctional morphology of ECs in the DA (Venus, grey) of a 0.5% DMSO, a 15 μM/0.5% DMSO Nifedipine and a 25 μM/0.5% DMSO Nifedipine treated embryo at 4 dpf. Treatment from 2 to 4 dpf. Scale bar = 10 μm. **d** Ratio-metric FRET values in junctions of 0.5% DMSO ($n = 42$ junctional ROIs from $n = 5$ embryos), 15 μM Nifedipine ($n = 54$ junctional ROIs from $n = 6$ embryos) and 25 μM Nifedipine ($n = 59$ junctional ROIs from $n = 6$ embryos) treated embryos. Treatment was from 2 to 4 dpf. Error bars represent mean ± s.d.; DMSO – 15 μM Nifedipine **p = 0.0068, DMSO – 25 μM Nifedipine ****p < 0.0001, from unpaired two-sided Mann–Whitney test.**e** Junctional linearity index in 0.5% DMSO treated controls ($n = 5$) versus 25 μM Nifedipine-treated embryos ($n = 6$). Each data point represents a junction, $n = 18$ control junctions measured and $n = 18$ junctions from Nifedipine-treated embryos. Error bars represent mean ± s.d.; not significant (ns) $p = 0.8147$, from unpaired two-sided Mann–Whitney test. **f** Ratio-metric FRET values quantified from 1% DMSO-treated controls ($n = 39$ junctional ROIs from $n = 6$ embryos), BAPTA-AM-treated (100 μM for 1 h, $n = 58$ junctional ROIs from $n = 8$ embryos), Nifedipine-treated (50 μM for 30 min, $n = 27$ junctional ROIs from $n = 5$ embryos) and Nifedipine + Y-27632-treated embryos ($n = 47$ junctional ROIs from $n = 8$ embryos). Error bars represent mean ± s.d.; 1% DMSO – 100 μM BAPTA-AM *p = 0.0498; 1% DMSO – 50 μM Nifedipine ****p < 0.0001; 50 μM Nifedipine - 50 μM Nifedipine + 50 μM Y-27632 ***p = 0.0008, from unpaired two-sided $t$ test

double mutants[30] and we observed single-MO phenotypes consistent with the reported, milder, single-mutant phenotypes[30]. In this context, we observed a clear morphological disruption of junctional maturation between 2–3 dpf (Fig. 4a, b) and an increased number of ECs in the DA (Fig. 4c). Interestingly, we observed increased VE-cadherin tension in these immature junctions (Fig. 4e, f). The downstream effector of Vegfr2 signaling in developing zebrafish arteries is Erk[31] and so to independently validate the observed changes, we turned to inhibition of Erk signaling. Treatment with the MEK inhibitor SL327, which blocks Erk phosphorylation in arteries[31], from 20 h post fertilisation (hpf) to 3 dpf also resulted in reduced linearity and increased VE-cadherin tension at 3 dpf (Fig. 4g–i). Interestingly, this treatment resulted in a milder effect on arterial angiogenesis when compared to the *kdr/kdrl* double morphants (Supplementary Fig. 3a,b), yet still impacted upon junctional maturation and VE-cadherin tension. Importantly, increased tension upon Vegfr2 depletion required acto-myosin activity, since Y-27632 treatment of double morphants lowered VE-cadherin tension (Fig. 4j). Overall, manipulating the normal process of maturation by reducing Vegfr2 or downstream Erk signaling during development leads to disorganized junctions that display increased tension through VE-cadherin. These observations further confirm the utility of the TS line to analyse

sub-cellular changes in functional vessels in vivo; both increases and decreases in intra-molecular tension across VE-cadherin.

**Acute loss of flow does not alter tension upon VE-cadherin.** The most well-studied forces that regulate vascular development and disease are those generated by blood flow. Lumenal flow generates shear stress and pressure, which control EC state and signaling[4]. To study if loss of blood flow might affect VE-cadherin tension, we first took advantage of the most commonly used loss-of-flow model in zebrafish, by injecting *silent heart* (*sih*) targeting MOs[32]. However, we found this model unsuitable since the ECs of the DA were severely disrupted and junctions failed to form without initial establishment of embryonic blood flow (Supplementary Fig. 4a). Alternatively, we blocked cardiac contraction after the arterial network had formed under normal flow conditions. We applied a high dose (800 mg/ml) of the anesthetsic tricaine (3-amino benzoic acid ethyl ester) to 3 dpf embryos, sufficient to stop contraction and thus flow. After 3 h without flow, ratio-metric FRET imaging revealed that tension through VE-cadherin was unchanged (Supplementary Fig. 4b, c). Furthermore, we surgically disconnected the outflow tract of 3 dpf embryos to stop blood flow and again VE-cadherin tension did not significantly change compared with control siblings

(Supplementary Fig. 4d–e). Finally, 50 mM 2,3-butanedione monoxime (BDM, a myosin ATPase inhibitor) at 2 dpf, a stage when VE-cadherin is under higher tension, did not alter VE-cadherin tension (Supplementary Fig. 4f–g). We have reported that VE-cadherin tension decreases from 2 to 4 dpf, as flow patterns have been reported to change[27, 33], a phenomenon that was also observed after induction of shear stress over an EC monolayer[5]. The failure to detect changes in VE-cadherin tension when flow was acutely disrupted might indicate changes that are below the detection limit using this TS system. Alternatively, the ongoing role of flow upon junctions of the DA may represent only a minor contribution relative to the changes induced over a longer developmental period by cellular rearrangements regulated by acto-myosin contractile forces.

**Calcium signaling maintains low VE-cadherin tension.** Endothelial calcium signaling is required for ongoing vascular morphogenesis in zebrafish[34] and is important for cellular homeostasis[35–37] and acto-myosin function[38, 39]. We therefore asked if the levels of intra-cellular calcium contribute to arterial maturation. We first assessed calcium levels in the DA from 2 dpf to 4 dpf using a Tg(fli1a:GALFF) driver combined with UAS:RFP and UAS:GCaMP5G (a calcium reporter[40, 41]). We found that during DA maturation, intra-cellular calcium increases, measured as the normalized index of GCaMP5G/RFP (Fig. 5a, b). To test if this increase in calcium plays a role in junctional maturation, we reduced EC intra-cellular calcium by treating embryos with the calcium channel antagonist Nifedipine[42–46]. Nifedipine treatment from 2 to 4 dpf resulted in a significant, concentration dependent, increase in VE-cadherin tension (Fig. 5c, d). Interestingly, these treatments did not affect junctional linearity (Fig. 5e). We further found that a higher dose of Nifedipine (50 µM) for 30 min at 3 dpf resulted in an immediate increase in VE-cadherin tension, which was prevented when ROCK was inhibited with Y-27632 (Fig. 5f). In addition, independent treatment with the intra-cellular calcium chelator BAPTA-AM in a pulsed experiment also increased VE-cadherin tension rapidly, similar to Nifedipine (Fig. 5f). These results demonstrate that calcium signaling in the DA increases during vessel maturation and is required for reduced tension through VE-cadherin. Importantly, the observation that calcium signaling inhibition leads rapidly to increased tension suggests that calcium plays an active, ongoing role to maintain low tension across junctional VE-cadherin during artery maturation.

## Discussion
This study shows that TS technology is applicable in a complex vertebrate vasculature. To our knowledge, this strain represents the first functionally tested, tissue restricted TS system demonstrated to measure tensile changes in a vertebrate. Hence, we expand on previous overexpression of an α-actinin-TS in frogs[47] and highlight the potential for translation of tension biosensors to other vertebrates, tissues, cellular structures and disease models. The line provides a tool to visualize VE-cadherin dynamics in great detail. Importantly, changes in energy transfer across zebrafish VE-cadherin-TS proteins is a reliable proxy for acto-myosin controlled tension through VE-cadherin. Hence, this line can be applied to study mechanical changes during vascular morphogenesis, in mutant models for cardio-vascular diseases or using pharmacological approaches.

Using VE-cadherin-TS as a junctional marker, we identified a correlation between changes in junctional morphology and decreased tension across VE-cadherin as arterial junctions mature in vivo, similar to what has been reported when flow was applied to cultured cells in vitro[5]. Over time, aortic EC junctions become

more linear and align with the direction of blood flow, resulting in more elongated ECs and reduced tension through VE-cadherin. We also saw that when Vegfr2 or Erk signaling is reduced, junctions fail to linearize, remain disorganized (immature) and remain under higher VE-cadherin tension. These mechanical changes probably relate to the dynamic state of junctions during vessel morphogenesis, which has been previously shown to be regulated by Vegfr-signaling levels[29]. Interestingly, between 2 and 3 dpf, this process of cellular rearrangement during artery maturation has been correlated with changes in flow patterns and is thought to be regulated by flow[27]. Nevertheless, we found that a complete loss of blood flow for a period of 3 h at 2 and 3 dpf has no influence on VE-cadherin tension. This may indicate that the mechanical changes controlling EC junctional maturation, if regulated by flow, become established over a longer developmental timeframe during maturation and that flow plays little role in maintenance of tension through VE-cadherin. Future analyses in mutants models such as the recently described endoglin-deficient model[27], in which the dorsal aorta fails to mature, resulting in patterned flow defects, may help to reveal how tension changes observed across VE-cadherin contribute to these stereotypical cellular rearrangements during arterial maturation.

We further identify an increase in intra-cellular calcium in maturing ECs, which correlates with the process of DA maturation from 2 to 4 dpf. We find that acutely reducing calcium signaling leads to immediate increases in VE-cadherin tension. Hence, unlike flow, calcium signaling maintains the junctional mechanical state as measured at the level of VE-cadherin tension. In zebrafish, the initiation of blood flow triggers endothelial calcium signaling, which is required for ongoing vascular morphogenesis[34]. Likewise, changes in flow conditions that occur during maturation[27, 33] could open endothelial calcium channels, thereby increasing intra-cellular calcium over time. Inside the cells, calcium probably facilitates reduced VE-cadherin tension by modulating the distribution of acto-myosin tension away from VE-cadherin at cell–cell junctions. Our work hence identifies a number of factors that contribute to tension changes across VE-cadherin during arterial maturation. Our data suggests that as the DA matures changes in junctional morphology, cellular orientation, calcium signaling levels, and contractile tension across VE-cadherin define the maturation process. Importantly, these studies of DA maturation demonstrate the utility of the TS model in one specific setting, and highlight the promise of this approach more broadly.

It is important to note several observations that should influence similar, future studies. The use of FRET imaging in vivo is technically challenging and could produce spurious biological conclusions if not carefully controlled. We have used well-defined controls and find that both technical imaging controls and biological controls are needed to ensure confidence. Specifically, to demonstrate that TS measurements reflect biological changes rather than imaging modality, we here used both ratio-metric and lifetime FRET imaging for key observations (e.g., tension changes in wild-type embryos during maturation and upon Y-27632 treatment). We also used multiple related genetic and chemical treatments to validate key observations (e.g., Y-27632 and Cdc42 inhibition; kdr/kdrl MO knockdown and SL327 treatment; Nifedipine and BAPTA-AM treatment; Tricaine, BDM and surgical blockage of flow). Furthermore, Y-27632 treatment served as an ideal negative control setting for treatments that caused increased tension through VE-cadherin (e.g., Vegfr2 depletion and Nifedipine treatment). More difficult to use in vivo was the VE-cadherin TL control construct that produced VE-cadherin lacking the β-catenin-binding domain and therefore unable to bind to the acto-myosin cytoskeleton. We found that junctional TL did

report higher FRET measurements (i.e., less tension); however, use of the construct was compromised because in many cases it mislocalized in the cell. This is likely due to the β-catenin-binding domain being required for efficient VE-cadherin localization, as supported by recent studies in *Drosophila*[48]. Thus, it is possible that such negative control constructs are unreliable and we suggest chemical inhibition of endogenous acto-myosin contractility serves as a more reliable negative control. We further noted above that FRET efficiency changes measured by lifetime were limited for this particular TS. We expect that ongoing optimization of TS modules with different force sensitivities[11, 13, 14] as well as improved sensitivity in microscopy in the future will allow us to improve this and to quantify intra-molecular tension and contractile force with increasing sensitivity. Finally, by analyzing VE-cadherin-TS on a VE-cadherin-mutant background and expressing it from a large BAC clone that appears to recapitulate normal gene regulation, we found that the TS protein could functionally compensate for loss of endogenous VE-cadherin. In the homozygous mutant context, we also observed VE-cadherin-TS expression at higher levels than in the presence of wild-type VE-cadherin, indicating accurate cellular regulation of VE-cadherin levels (Supplementary Fig. 2b). This increased expression is expected to improve image resolution, sensitivity, and more accurately reflect endogenous protein levels. We conclude that replacing the endogenous protein in such a manner is an ideal approach, superior to forced overexpression and recommended for future studies. Overall, the approaches and findings presented here highlight the utility of TS approaches in vivo and will help to support future studies into the physical forces controlling functional and developing tissues.

## Methods

**Zebrafish husbandry and strains**. All animal work adhered to the guidelines of the animal ethics committee at the University of Queensland. Published transgenic lines and mutants used for this work are *Tg(fli1a:GALFF)*[ubs3 49], *Tg(10.5xUAS: GCaMP5G)*[uq2Tg 41], *Tg(10xUAS:Venus)*[uq8bh50] and *cdh5(ve-cad)*[ubs8 21]. *Tg(5xUAS: RFP)*[nkuasrfp1a] was kindly provided by the Bakkers laboratory. All embryos were analysed before sexual maturity thus sex-specific differences can be excluded.

**VE-cadherin TS and VE-cadherin TL cloning and BAC recombineering**. To construct *ve-cadherin Tension Sensor* (*TS*) and *ve-cadherin Tail Less* (*TL*) cDNAs we initially replaced eGFP in pEGFP-N1 by the TS module sequence[8] using XhoI and NotI with In-fusion HD cloning technology. We generated two plasmids using this approach, one harboring a TS module without a stop codon (pN1-TS, for *ve-cadherin TS*) and one harboring a TS module with a stop codon (pN1-TS-stop, for *ve-cadherin TL*). We subsequently amplified the *ve-cadherin* "head" and "tail" domains of zebrafish *ve-cadherin* using a full-length clone (MGC:194987) as a template. The "head" domain includes the extracellular, transmembrane and *p120*-catenin domains of zebrafish *ve-cadherin* (Fig. 1b and Supplementary Fig. 1a). The "tail" includes the β-catenin-binding domain (Fig. 1b). The β-catenin-binding domain is absent in the VE-cadherin Tail Less negative control (Supplementary Fig. 2c).

Primers used for "head" and "tail" domain amplification: *ve-cad*-head-Forward:5′-**GGACTCA GATCTCGAG**CGCCACCATGATGAAACAGTGTGCC AG-3′

(infusion homology arm = bold, kozak sequence underlined)

*ve-cad*-head-Reverse: 5′-**TGCTCACCATCTCGAG**ATCTCGATCACGATCTG CC-3′

(infusion homology arm = bold)

*ve-cad*-tail-Forward: 5′-**CTGTACAAGGCGGCCGC**TGGCATTCCCTATGAC ACATTAC-3′

(infusion homology arm = bold)

*ve-cad*-tail-Reverse: 5′-**TCTAGAGTC GCGGCCGC**TCAGTAGGAGCTATCC GAATC-3′

(infusion homology arm = bold)

For *ve-cadherin TS* cDNA, the "head" domain was placed upstream of the TS module and the "tail" sequence was placed downstream of the TS module by infusion cloning at the XhoI and NotI restriction sites in pN1-TS making pN1-ve-cadTS. For *ve-cadherin TL* cDNA the "head" domain was placed upstream of the TS module by infusion cloning at the XhoI restriction site in pN1-TS-stop, making pN1-ve-cadTL. To generate inserts for BAC recombineering we amplified *ve-cadherin TS* (from pN1-ve-cadTS) and *ve-cadherin TL* (from pN1-ve-cadTL) including a Kanamycin resistance cassette using the following primers:

*ve-cad*-BAC insert-Forward: 5′-**ctacaaggcatgaattatcatacagatatttgttttgttgttttc tgcag**ATGATGAAACAGTGTGCCAGAAGG-3′ (homology arm = bold)

*kanamycin*-Reverse: 5′-**ACAGCAACCCTGAAGACAGGCTCAGTCATCT GCCTTCTGGCACACTGTTT**TCAGAAGAACTCGTCAAGAAGGCG-3′ (homology arm = bold)

We replaced the first two amino acids (ATGATG) of *ve-cadherin* on the CH73-357K2 BAC clone[15] with *ve-cadherin-TS-kanamycin* and *ve-cadherin-TL-kanamycin* inserts generating two BAC clones vecad:vecadTS and *ve-cad:ve-cadTL* (Supplementary Fig. 2c). Two Tol2 LTRs flanking an ampicillin resistance cassette were placed in the BAC vector backbone using RedET assisted recombination as previously described[15].

**Gateway cloning**. Teal and Venus were cloned into the Gateway pME vector (pDON-221) using Gateway technology[51].

Primers used:- pME-*Teal and Venus*-Forward: 5′-**GGGGACAAGTTTGTACA AAAAAGCAGGCT**atggtgagcaagggcgaggag-3′ (gateway homology arm = bold) - pME-*Teal and Venus*-Reverse: 5′-**GGGGACCACTTTGTACAAGAAAGCTG GGTA**ctacttgtacagctcgtccatg-3′ (gateway homology arm = bold)

Subsequently a Gateway LR reaction was performed combining a p5E-10xUAS, pME-Teal or pME-Venus and p5E-polyA placing the final 10xUAS:Teal and 10xUAS:Venus sequence into pDestTol2pA2AC (containing the α-crystallin promoter driving GFP in the zebrafish lens).

*fli1ep:lifeact-mCherry* plasmid DNA was a kind gift from Holger Gerhardt and Li-Kun Phng.

**Transgenesis**. To establish *Tg(ve-cad:ve-cadTS)*[uq11bh] 1 nl of purified BAC DNA was injected at 100 ng/μl into single-cell stage embryos together with *tol2* transposase mRNA (25 ng/μl). Embryos expressing VE-cadherin-TS mosaically were selected and raised to adulthood and screened for germline transmission to generate a stable transgenic line.

To establish *Tg(fli1ep:lifeact-mCherry)*[uq12bh] and *Tg(10xUAS:Teal)*[uq13bh] 1 nl of circular plasmid DNA was injected at 50 ng/μl into single cell stage embryos together with *tol2* transposase mRNA (25 ng/μl). Embryos expressing either β-crystallin:mKate2 (in backbone of *fli1ep:lifeact-mCherry* DNA) or α-crystallin:GFP (in backbone of *10xUAS:Teal* DNA) in the eyes were raised to adulthood and screened for germline transmission to generate stable transgenic lines.

**Experimental design VE-cadherin-TS rescue experiment**. At 1 day post fertilization (dpf), when homozygous mutants are phenotypically indistinguishable from siblings, embryos were separated in two populations; *Tg(ve-cad:ve-cadTS)*-positive and *Tg(ve-cad:ve-cadTS)*-negative embryos. We subsequently quantified the number of embryos that represented either a wild-type or *cdh5*[ubs8−/−] mutant phenotype at 2 dpf (Fig. 2b). The *Tg(ve-cad:ve-cadTS)*-positive embryos were raised to adulthood and we verified that *Tg(ve-cad:ve-cadTS)*[+/−], *cdh5*[ubs8−/−] mutant fish are adult viable as crossing these to non-transgenic *cdh5*[ubs8+/−] carriers resulted in 52% phenotypic mutants in *Tg(ve-cad:ve-cadTS)* negative clutches using the same experimental design (Fig. 2d).

**Chemical treatments**. For Y-27632 (ROCK inhibitor) treatments embryos were incubated in E3 medium containing 45 μM Y-27632 in 1% DMSO or 1% DMSO only as a negative control for 5 h at 3 dpf.

For ML-141 (Cdc42 inhibitor) embryos were incubated in E3 medium containing 50 μM ML-141 in 0.5% DMSO or 0.5% DMSO in E3 as a negative control for 5 h at 3 dpf.

Nifedipine (calcium channel inhibitor) was added at 15 and 25 μM in 0.5% DMSO in E3 medium from 2 dpf to 4 dpf. 0.5% DMSO in E3 was applied as a negative control over the same time course. For acute treatments at higher dose, Nifedipine was added at 50 μM in 1% DMSO with 1% DMSO only as a control. Embryos were treated for 30 min at 3 dpf. Embryos were treated with the cell-permeable calcium chelator BAPTA-AM (100 μM) in 1% DMSO with 1% DMSO only as a control for 1 h.

For SL327 (MEK inhibitor) treatments embryos were incubated in 4 uM SL327 in 1%DMSO or in 1% DMSO as a negative control from 20 hpf till 3 dpf. For imaging at 3 dpf embryos with intact blood circulation were selected for ratiometric FRET imaging.

To stop blood flow, we applied Tricaine at 800 mg/ml in E3 medium to 3 dpf embryos, which immediately stopped cardiac contraction and we used the myosin ATPase inhibitor 2,3-butanedione monoxime (BDM) at 50 mM to stop contraction at 2 dpf.

**Morpholino injections**. To prohibit cardiac contraction, we made use of a previously published morpholino targeting *silent heart/tnnt2a*[32]. To reduce Vegfa signaling, we injected published morpholinos against *kdr* and *kdrl* (1 ng each)[52, 53]. We optimized the *kdr* and *kdrl* morpholino reagents to achieve a partial loss of Vegfr2 function since complete loss of function as seen in *kdr/kdrl* double mutants[30], would result in cardiac defects and loss of blood circulation. At 2 dpf, we selected morphant embryos that had not formed intersegmental vessels (ISVs) but did have intact cardiac function and blood circulation (Supplementary Fig. 3a and Supplementary Movie 1).

**Genotyping cdh5ubs8 allele in Tg(ve-cad:ve-cadTS) animals.** To distinguish the wild-type *ve-cadherin* sequence within the BAC transgenic insertion from the endogenous *ve-cadherin* locus in *Tg(ve-cad:ve-cadTS)* embryos analysed in Supplementary Fig. 2b, we designed primers to specifically amplify a genomic region flanking the mutation site of *cdh5ubs8*. The PCR-amplified products were subsequently digested for 1 h at 60 °C using BsaBI, which cuts once in the wild-type allele and twice in the mutant (*ubs8*).

Primers used: *ubs8* in TS – Forward: 5′- ACAGTGTGTTTGCATCATTG-3′
*ubs8* in TS – Reverse: 5′-ACAGTCTTGGTGTTACCATTGGG-3′.

**Post-acquisition analysis of ratio-metric FRET values.** All FRET imaging was performed on Zeiss 710 FCS confocal microscopes using line-sequential scanning and GaAsP high sensitivity detectors.

Image settings were optimized for *Tg(ve-cad:ve-cadTS)* expression for every imaging session and then *Tg(ve-cad:GALFF); Tg(10xUAS:Teal)-* and *Tg(ve-cad: GALFF); Tg(10xUAS:Venus)*-positive embryos were imaged with the same settings (Supplementary Fig 1c). We used a custom-written Matlab (MathWorks) script[54] to measure nonlinear Donor BT (DBT) and Acceptor BT (ABT) constants from *Tg (ve-cad:GALFF); Tg(10xUAS:Teal)* and *Tg(ve-cad:GALFF); Tg(10xUAS:Venus)* images. All pixel intensities for both Teal and Venus in the images acquired from *Tg(ve-cad:ve-cadTS)* embryos that fell outside the confidence boundaries applied to measure DBT and ABT were excluded from the analysis by applying a MASK. As such, we processed only those RAW FRET-expressing pixels for which Donor and Acceptor intensities allowed the precise calculation of DBT and ABT constants. All processing to generate FRET index values was performed using ImageJ 1.47 (National Institutes of Health, USA) software on mean or median filter-adjusted RAW images. The final FRET index represents a BT-corrected FRET divided by Acceptor (Venus) intensity in each pixel (order of imaging processing is represented in Supplementary Fig. 1d).

**Junctional linearity index quantification.** Junctional linearity (Figs. 3b, 4d, h, and 5e) was quantified using ImageJ 1.47 (National Institutes of Health, USA) software. We generated maximum intensity projections of the Venus signal from *Tg(ve-cad: ve-cadTS)*-positive embryos. We projected one half of the dorsal aorta (DA) so that there would be no interference of junctional expression from junctions located at the other half. We selected only those junctions for which we could identify the vertices within the projection of the DA. Using the Straight-line tool, we measured the length in μm between the vertices. Then, using the Freehand-line tool, we measured the length of the junction in μm. The junctional linearity index was calculated by dividing the distance between junctional vertices over the length of the junction. The closer to 1 in the index, the straighter the junction.

**Quantification of junctional angles relative to flow.** Quantifications of the angle of junctions relative to blood flow (Fig. 3c) was performed using ImageJ 1.47 (National Institutes of Health, USA) software. We generated maximum intensity projections of the Venus signal from *Tg(ve-cad:ve-cadTS)*-positive embryos and selected only those junctions for which we could identify the vertices within the projection of the DA. We projected one half of the DA so that there would be no interference of junctional expression from junctions located at the other half. Using the Staight-line tool, a reference line was drawn through the middle of the vessel to indicate the direction of blood flow. For each junction, the Angle-tool was applied to draw the angle between the reference line of flow and the direction of the junction and measured angles were noted. The smaller the angle, the more the junction is aligned to flow.

**Quantifications of the number of ECs in the dorsal aorta.** To quantify the number of ECs in the DA (Fig. 4c), we used the junctional expression of Venus from *Tg(ve-cad:ve-cadTS)* to identify the cell–cell junctions and thus select individual cells. We generated maximum intensity projections of a ~213 μm long fragment of the DA. We projected one half of the DA so that there would be no interference of junctional expression from junctions located at the other half. The Multipoint-tool in ImageJ 1.47 (National Institutes of Health, USA) was applied to mark each cell and these were saved as ROIs.

**Post-acquisition analysis of GCaMP5G expression.** Imaging of *Tg(fli1a:GALFF); Tg(10.5xUAS:GCaMP5G); Tg(5xUAS:RFP)* embryos was performed on a Zeiss 710 FCS confocal microscopes using GaAsP detectors. The same embryos were imaged at 2 and 4 dpf, and the image settings were kept constant. All images were filter mean = 1 adjusted in ImageJ 1.47 (National Institutes of Health, USA) and the RFP expression was used to make a binary MASK of the endothelial expression. The GCaMP5G expression was subsequently divided over the RFP expression present within the MASK region to generate the final GCaMP5G/RFP index. The index measurements were collected from 5 slices of the dorsal wall of the DA in each embryo. The average of the five slices was noted as the final GCaMP5G/RFP index value (Fig. 5b).

**Correlation analysis of Venus intensity and ratio-metric FRET.** For pixel-matched correlation analysis, we used the line-tool in ImageJ 1.47 (National

Institutes of Health, USA) to draw lines across the middle of the junctions and then plotted the intensity profiles of both Venus fluorescence intensities and ratio-metric FRET values present in the pixels across these lines. These profiles are shown in Supplementary Fig. 1e–h. Correlation plots of these pixel-matched values are shown in Supplementary Fig. 1e′–h′. In addition to the pixel-matched analysis, a grouped analysis was performed of all pixels within all junctions ROIs (n = 52) from different fish (n = 10) at 4 dpf (Supplementary Fig. 1i). First junctional pixels were isolated using a binary MASK in ImageJ 1.47 (National Institutes of Health, USA) based on Venus fluorescence. Subsequently, all pixels were grouped based on increments of Venus fluorescence intensity levels of 500 (group1: 7750-8250, group2: 8250-9750, group3: 9750-10250 …… and group28: 35250–35750). The mean FRET index and S.E.M of pixels that belong to a Venus intensity window were calculated and plotted for each Venus intensity group as a function of the Venus intensity.

**Grouped analysis of pixel distribution across intensity groups.** To compare the fluorescence intensity levels of Venus in junctions from heterozygous *ubs8*[+/−] and homozygous *ubs8*[−/−]-mutant embryos (all *Tg(ve-cad:ve-cadTS)*[+/−]) (Supplementary Fig. 2b), we quantified the number of junctional pixels within groups separated by increments of Venus intensity of 1000; group1: 6000–7000 (mean = 6000), group2: 7000–8000 (mean = 7500), group3: 8000–9000 (mean = 8500) …… group41: 47000–48000 (mean = 47500). Junctional pixels were selected by making a binary MASK in ImageJ 1.47 (National Institutes of Health, USA) based on Venus fluorescence. The number of pixels per group was plotted against the mean Venus fluorescence intensity for that group (e.g., mean = 6000, 7500, 8500, etc).

**FLIM-FRET imaging and processing.** The data for FLIM (Figs. 2h, i, and 3f, g) was acquired using an HC PL APO CS2 40 × 1.1 NA water objective on an inverted DMI 6000 SP8 confocal microscope (Leica Microsystems). The excitation source was a Ti:Sapphire femtosecond laser cavity (Chameleon Vision II, Coherent), operating at 80 MHz and tuned to a wavelength of 870 nm for Teal excitation. Intensity was recorded with HyD detector (Leica Microsystems) in the non-descanned (NDD) position (520/40 nm). The FLIM data were recorded using a Picoharp 300 TCSPC system with Symphotime software (Picoquant). Images of 512 × 512 pixels were acquired with a line rate of 600 Hz. The pixel dwell time was 0.8 μs, and images were integrated until 200-300 photons per pixel were acquired. All processing of the RAW FLIM data was performed using FLIMfit software[55] version 5.0.1 (kindly provided by Dr Sean Warren). First, to select reads from the junctions and exclude background counts, junctional regions were segmented. Subsequently the segmented images were smoothed using a 3 × 3 pixel kernel and the mean lifetime per junctional region was calculated from a single exponential fit to the fluorescence decay data using a Maximum Likelihood estimator. The pixels in each junction were averaged to give a single lifetime value (one data point = one junction). Pixels below the background levels of intensity are set to black in the color-mapped images and within junctions blue is low lifetime and red is high lifetime. The mean FRET efficiencies (E) measured by FLIM were calculated by dividing the mean Teal lifetime values by the published lifetime of Teal[56] (2.65 nano-sec) in the following equation: E = 1-(mean Teal lifetime/2.65).

**Statistical analysis.** Prism software (GraphPad) was used for all statistical analysis (*t* tests and Mann–Whitney-tests). D'Agostino–Pearson test was applied to test normal distribution of the data points. When the data were normally distributed a Student's *t* test was used for comparison of two means. When the data did not follow a normal distribution, the Mann–Whitney test was used for comparison of two means. The threshold for significance was taken as p < 0.05 and all the data are represented as mean ± s.d. For each experimental comparison, embryos were randomly distributed and sibling matched, therefore each embryo can be considered as a biological replicate. No statistical methods were used to predetermine sample sizes, but our sample sizes are comparable to those reported recently for analysis of junctional dynamics in zebrafish[27, 57].

**Data availability statement.** All the relevant data supporting the findings are available from the corresponding author on reasonable request.

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

## Acknowledgements

A.K.L. was supported by a UQ Postdoctoral Fellowship, B.M.H. by an NHMRC/National Heart Foundation Career Development Fellowship (1083811) and A.S.Y. by an NHMRC Research Fellowship (1044041). This research was supported by an ARC Discovery Project grant (DP150014119) and NHMRC grants (1067405, 1037320). We thank Kylie Georgas for design of graphics in the manuscript. We thank Dr. Sean Warren for assistance with FLIM acquisition and kindly providing us with FLIMfit software[55] for lifetime analysis. We thank H. Gerhardt and L-K. Phng for kindly providing DNA constructs that we used to generate the *Tg(fli1ep:lifeact-mCherry)*[uq12bh] strain. We thank E. Scott for providing the *Tg(10.5xUAS:GCaMP5G)*[uq2Tg] stain[41]. Imaging was performed in the Australian Cancer Research Foundation's Dynamic Imaging Facility at IMB (established with the generous support of the ACRF), in the Queensland Brain Institute's Advanced Microscopy Facility and generously supported by ARC LIEF LE130100078 and in the Microscope Facility of the Garvan Institute of Medical Research.

## Author contributions

A.K.L., B.M.H., A.S.Y., G.A.G., and M.A.S.: Conceived the idea and directed the work. A.K.L., B.M.H., A.S.Y., and G.A.G.: Designed the experiments; A.K.L., G.A.G., S.B., and

S.P.: Performed the experiments, D.H., and W.E.H.: Provided facilities and assisted in FLIM imaging, D.E.C., H-G.B., M.A., K.A.S., and M.A.S.: Shared unpublished reagents, A.K.L., B.M.H., and A.S.Y.: Wrote the manuscript.

## Additional information

**Competing interests:** The authors declare no competing financial interests.

