## [Peer Review File · Nature Communications]

Reviewers' comments:

Reviewer #1, expert in zebrafish vascular development (Remarks to the Author):

In this paper, the authors generate a FRET based VE-cadherin tension biosensor transgenic zebrafish line and use it to analyze changes in VE-cadherin tension as the dorsal aorta (DA) matures and endothelial cell (EC) junctions become more linear. They also show that when Vegfr2 signaling is reduced, junctions fail to linearise and remain under higher VE-cadherin tension. These data confirm in vitro data that VE-cad can act as a mechanosensor along EC junctions and thus the VE-cad tension sensor tool will most be valuable to examine the potential role and dynamics of physical forces during endothelial development. The authors go on to identify an increase in intracellular calcium in maturing EC's, which contributes to the progressive reduction of VE-cadherin tension over time.

The observations and findings are very interesting and even though the underlying molecular mechanisms remain mostly uncharacterized, the technical novelty of this work is sufficient for this initial paper to be published in Nat. Comm. after addressing a few points.

Questions:

- If VE-cad TS can act as an endogenous VE-cad, is there an effect of its expression in wild-type animals (not VE-cad mutants)?

- Is there a functional significance of the decrease in tension across VE-cad as the embryos develop (2-4 dpf)? Or is this more of a consequence of reorganization and stabilization of actin dynamics?

- The junctions in Nifedipine (calcium channel inhibitor) treated fish appear to be normal. Does it mean that junction formation and calcium signaling have independent roles in reducing VE-cadherin tension during DA maturation?

- The authors use silent heart MO to examine the effects of blood flow on VE-cadherin tension, but because of the defects on cell-cell junction formation they couldn't. How about using tricaine or another anesthetic to modulate cardiac contraction once circulation is established?

- abstract: 2nd sentence, should it say: 'acto-myosin network'?

- page 9: all allele numbers should be included

Reviewer #2, expert in molecular tension sensors (Remarks to the Author):

This manuscript by Legendijk et al. entitled "Live imaging molecular changes in junctional tension upon VE-cadherin in zebrafish" describes the generation and analysis of a VE-cadherin tension sensor zebrafish strain. After validation of protein functionality, the authors demonstrate that VE-cadherin bears mechanical forces in endothelial junctions in an actomyosin-dependent fashion, as described in cell culture experiments before (Conway et al., 2013). In addition, the presented experiments reveal changes in VE-cadherin tension during dorsal aorta maturation that appear to depend on Vegfr2 and calcium signaling. The manuscript is well-written and the experimental observations are interesting. However, experimental controls are missing and it is insufficiently explained how the data have been statistically evaluated. There are also some minor issues that should be corrected. Please see my comments below.

Major issues:

- 1) The statistical evaluation of FRET data should be explained in more detail. It is unclear what a single data point in the FRET plots actually stands for (e.g., Fig. 2f, h; Fig.3i, f; Fig.4d, e; Supplementary Fig. 1h; Supplementary Fig. 2d). Is this the average of all junctions from one animal, or one junction? What's the 'n'? How many experimental days are included in these data sets? It will be hard for the reader to evaluate the data without knowing what the data points actually represent and I strongly encourage the authors to be as clear as possible by either writing a separate section on this issue in the methods part, or by including this information directly in the figure legends.
- 2) By generating the VE-cadherin-TL construct, the authors have produced a very useful control to evaluate potential unspecific (i.e. tension independent) effects on FRET. Unfortunately, the control is only used once (Fig.2f) but is missing from all other FRET experiments. How can the authors exclude unspecific effects on FRET in these experiments? I find the lack of a negative control problematic for a number of reasons: First, the authors use ratiometric FRET imaging to determine a FRET index, which is sensitive to the instrumental settings. As expected, the FRET indices vary between experiments and therefore can hardly be compared with each other (e.g. the FRET index of VE-cadherin-TS is about 0.1 in Fig.2f, 0.3 in Fig.2h, 0.45 in Fig.4e; VE-cadherin-TL is about 0.25 in Fig.2f). Second, the authors lack an intermolecular FRET control. Third, Fig. 2e indicates quite a variation of FRET in the VE-cadherin-TL sample, consistent with Fig. 2f which also shows a larger spread in the VE-cadherin-TL data as compared to the VE-cadherin-TS. This may not be a problem per se, but it shows that tension-independent effects are present and should be controlled for. Finally, some FRET differences are rather small (e.g. Fig. 3i, j). Thus, it would significantly strengthen the paper if these experiments were more rigorously controlled. I suggest to include VE-cadherin TL measurements in the experiments described in Fig.2h, Fig.3d, i, j and Fig. 4d, e.

Minor issues:

- 1) The references should be checked again. In at least one instance the provided reference does not make much sense in the given context (line 53, ref.7).
- 2) The numbers of analyzed animals in Fig. 2c, d should be mentioned.
- 3) In line 472, it should read (j) not (i).

Reviewer #3, expert in cell-cell adhesion, in vivo imaging (Remarks to the Author):

This paper from Lagendijk and colleagues describes the introduction of a VE-cadherin tension sensor into the zebrafish vasculature, validation of the construct in vivo, and initial assessments of changes to junctional tension during vascular development. The strategy has been previously used in cell culture but this is the first report of a junctional tension sensor in a developing vertebrate. The authors employed an appropriate and well-controlled approach for assessing the FRET-based sensor. The sensor is expressed at endogenous VE-cadherin levels and can rescue VE-cadherin mutants. They show that tension detection requires the sensor's beta-catenin binding site, requires actomyosin activity and is lessened as cell-cell contacts of vasculature smoothen out with development, all properties consistent with published in vitro analyses of the sensor. Providing new insights into junctional tension in vivo, they report evidence of requirements of Vegfr2 and calcium for the developmental reduction of tension. The main strength of the paper is the report of a useful tool for studying junctional tension in vivo, a topic of interest in field of tissue morphogenesis. My main concern is the paper's limited conceptual advances, but this concern is typical for a technique-focused paper.

I also have two specific concerns about the conceptual advances.

1. Evidence for the specificity of the applied vegfr2 morpholinos should be provided. Can the effect be rescued with morpholino-resistant mRNA, or can it be seen with mutants?

2. Evidence for the specificity of the applied calcium channel blocker should be provided. Can the effect be seen with a distinct calcium channel blocker?

Detailed Response to Reviewers' comments:

Reviewer #1: In this paper, the authors generate a FRET based VE-cadherin tension biosensor transgenic zebrafish line and use it to analyze changes in VE-cadherin tension as the dorsal aorta (DA) matures and endothelial cell (EC) junctions become more linear. They also show that when Vegfr2 signaling is reduced, junctions fail to linearise and remain under higher VE-cadherin tension. These data confirm in vitro data that VE-cad can act as a mechanosensor along EC junctions and thus the VE-cad tension sensor tool will most be valuable to examine the potential role and dynamics of physical forces during endothelial development. The authors go on to identify an increase in intra-cellular calcium in maturing EC's, which contributes to the progressive reduction of VE-cadherin tension over time.

The observations and findings are very interesting and even though the underlying molecular mechanisms remain mostly uncharacterized, the technical novelty of this work is sufficient for this initial paper to be published in Nat. Comm. after addressing a few points.

Questions:

1. If VE-cad TS can act as an endogenous VE-cad, is there an effect of its expression in wild-type animals (not VE-cad mutants)?

We thank the reviewer for this question. We agree that it is important to clearly establish the effect of VE-cadherin-TS protein expression in a wild-type animal since additional VE-cadherin could result in phenotypes. We now provide **new data in Supplementary Figure 2** to address this question. Based on our new data we conclude that expression of VE-cadherin-TS does not affect embryonic development or the viability of wild-type animals.

Specifically, *Tg(ve-cad:ve-cadTS)* transgenic animals carrying either one or two copies of the transgene are viable and fertile and we have not been able to identify any phenotypic consequences of the additional VE-cadherin expression. To date, we have been breeding *Tg(ve-cad:ve-cadTS)* in wild-type, *cdh5^{ubs8+/-}* heterozygous and *cdh5^{ubs8-/-}* homozygous mutant backgrounds successfully for 7, 6 and 7 generations respectively. We have now added images of adult *Tg(ve-cad:ve-cadTS)* fish that are genotypically wild-type (top) and *cdh5^{ubs8-/-}* mutant (bottom) in Supplementary Fig. 2a to demonstrate that these are morphologically indistinguishable. While we cannot exclude the possibility of subtle phenotypic consequences of the presence of the sensor, they are not detectable at a gross phenotypic level.

In addition, we have quantified the expression levels of VE-cadherin in heterozygous and homozygous mutants (all carrying a single copy of the TS transgene) and added this data in Supplementary Fig 2b. In this setting where

the endogenous copies of the VE-cadherin gene are progressively lost, we see that the levels of the TS protein adjust and increase. This indicates that the endothelial cells can regulate the levels of VE-cadherin at the junctions and that the BAC clone containing the sensor remains responsive to this dosage regulation. Hence, it is likely that the TS does not represent an overexpression scenario. Also, this justifies using the TS in the mutant background, which improves the detectable range of VE-cadherin-TS expression at the junctions.

We add the following text to page 6-7 to point this out:

*“We next examined the amount of VE-cadherin protein (by quantification of Venus fluorescence intensity) at junctions in heterozygous and mutant animals (all Tg(ve-cad:ve-cadTS)^{+/-}), and we found that the gene dosage of the wild-type VE-cadherin allele influences the amount of transgene derived VE-cadherin at the junctions. This was observed as higher Venus fluorescent intensities, at junctions in homozygous mutants (**Supplementary Fig. 2b**) This observation indicates that endothelial cells (ECs) actively regulate functional VE-cadherin levels at their adherens junctions and suggests that forced overexpression of TS constructs is unlikely to generate data reflecting the normal physiologically regulated conditions.”*

We have also added a new section in the discussion on page 15 to emphasize the advantages of imaging VE-cadherin-TS in the mutant background:

*“Finally, by analyzing VE-cadherin-TS on a VE-cadherin mutant background and expressing it from a large BAC clone that appears to recapitulate normal gene regulation, we found that the TS protein could functionally compensate for loss of endogenous VE-cadherin. In the homozygous mutant context, we also observed VE-cadherin-TS expression at higher levels than in the presence of wild-type VE-cadherin, indicating accurate cellular regulation of VE-cadherin levels (**Supplementary Fig. 2b**). This increased expression is expected to improve image resolution, sensitivity and more accurately reflect endogenous protein levels. We conclude that replacing the endogenous protein in such a manner is an ideal approach, superior to forced overexpression and recommended for future studies.”*

2. Is there a functional significance of the decrease in tension across VE-cad as the embryos develop (2-4 dpf)? Or is this more of a consequence of reorganization and stabilization of actin dynamics?

This is a difficult issue to address experimentally.

When we treat with Y-27632, which reliably reduces tension through VE-cadherin, this is not sufficient to induce maturation of the junctions. We treated embryos with a higher dose of Y-27632 (100µM) for 5 hours at 2 dpf and 3 dpf but did not measure any improvement in junctional linearity as a readout for maturation (not shown).

New data on Erk signaling inhibition shows that with relatively normal vascular development upon inhibitor (SL327) treatment, we see increased tension in visibly less mature junctions. This data is now included in **Figure 4** and combines with *kdr/kdrl* MO data. This supports the idea that more disorganized junctions display higher tension and this is dependent on acto-myosin contractility regulated by ROCK (**Fig. 4g**). This data is correlative and mechanistically, we have not been able to separate the cause of these changes at the level of VE-cadherin regulation or actin dynamics.

Disentangling actin dynamic contributions from decreased VE-cadherin tension more definitively represents a study in itself and we believe this is beyond the scope of the current paper.

3. The junctions in Nifedipine (calcium channel inhibitor) treated fish appear to be normal. Does it mean that junction formation and calcium signaling have independent roles in reducing VE-cadherin tension during DA maturation?

The reviewer is correct and this is an interesting point. Junctional maturation is not affected by low dosages of Nifedipine during arterial maturation. We have now quantified junctional linearisation of embryos treated with 25 μ M Nifedipine from 2 to 4 dpf compared to DMSO treated controls and we did not find a significant difference. This data has now been added **as new data in Figure 5 in panel e**. We interpret this to mean that calcium signaling has an ongoing role in maintaining mechanical tension levels through VE-cadherin rather than regulating the cellular rearrangements that occur during the maturation process.

To synthesise these findings we added the following text on page 12:

*“Interestingly, these treatments did not affect junctional linearity (**Fig. 5e**). We further found that a higher dose of Nifedipine (50 μ M) for 30 minutes at 3 dpf resulted in an immediate increase in VE-cadherin tension, which was prevented when ROCK was inhibited with Y-27632 (**Fig. 4j**). In addition, independent treatment with the intra-cellular calcium chelator BAPTA-AM in a pulsed experiment also increased VE-cadherin tension rapidly, similar to Nifedipine (**Fig. 4j**). These results demonstrate that calcium signaling in the DA increases during vessel maturation and is required for reduced tension through VE-cadherin. Importantly, the observation that calcium signaling inhibition leads rapidly to increased tension suggests that calcium plays an active, ongoing role to maintain low tension across junctional VE-cadherin during artery maturation.”*

4. The authors use silent heart MO to examine the effects of blood flow on VE-cadherin tension, but because of the defects on cell-cell junction formation they couldn't. How about using tricaine or another anesthetic to modulate cardiac contraction once circulation is established?

We thank the reviewer for this important suggestion. We agree that other methods can be tested to alter flow at later time points. We have now measured VE-cadherin tension in junctions of the dorsal aorta upon loss of blood flow at 2 dpf and 3 dpf using three independent methods. The results of these experiments have now been added **as new data in Supplemental Figure 4**.

First, as suggested by the reviewer we incubated *Tg(ve-cad:ve-cadTS)* embryos (n=10) at 3 dpf in a high dose of the anesthetic Tricaine (800mg/ml) which resulted in an immediate loss of cardiac contraction and thus blood flow. We then performed FRET imaging after 3 hours and quantified VE-cadherin tension compared to untreated control embryos (n=10). We did not observe a significant difference in tension across VE-cadherin upon a 3-hour absence of flow using this method (**Supplementary Fig. 4b-c**).

Secondly, and in addition to this chemical interference, we also established a method of ceasing blood flow manually by surgically disconnecting the heart from the vasculature using forceps at the position of the outflow tract in 3 dpf *Tg(ve-cad:ve-cadTS)* embryos. After 3 hours of blood flow loss the embryos were embedded and imaged simultaneously to untreated siblings. In accordance with the results obtained with high Tricaine, we did not measure significant changes in VE-cadherin tension when blood flow was lost using this surgical procedure (**Supplementary Fig. 4d-e**).

Finally, we were interested to analyse the effect of flow loss at 2 dpf, since VE-cadherin is under higher tension at this stage compared to 3 dpf which might make this a more suitable stage to detect changes in tension upon flow loss. We examined the effect of 2,3-butanedione monoxime (BDM) treatment in 2 dpf embryos. BDM is a skeletal-myosin ATPase inhibitor and is a commonly used reagent to interfere with cardiac contraction in zebrafish embryos¹⁻⁴. In accordance with the data we acquired at 3 dpf, we did not measure a significant change in tension through VE-cadherin 3 hours post BDM treatment (**Supplementary Fig. 4f-g**).

When flow is lost acutely during this time-window (either at 2 or 3 dpf) we do not detect significant changes in VE-cadherin tension. Together with previous data in cultured cells showing that VE-cadherin is under tension in no-flow conditions⁵ this suggests that a steady state of VE-cadherin tension is maintained in no flow conditions. However, tension changes upon acute flow loss might also be of a scale below the detectable limit of the TS system. More likely, tension changes may occur over longer timeframes than 3 hours as flow changes over time control cellular rearrangements to allow artery maturation⁶. Here, we show that these same cell shape and junctional maturation changes correlate with a decrease in tension over time (**in Fig.3**)

The following text is now included to synthesize these points:

On page 11:

“The failure to detect acute changes in VE-cadherin tension when flow was abruptly disrupted might indicate changes that are below the detection limit using this TS system. Alternatively, the ongoing role of flow upon junctions of the DA may represent only a minor contribution relative to the changes induced over time by cellular rearrangements regulated by acto-myosin contractile forces.”

And in the discussion on page 13:

“Interestingly, between 2 and 3 dpf this process of cellular rearrangement during artery maturation has been correlated with changes in flow patterns and is thought to be regulated by flow. Nevertheless, we found that a complete loss of blood flow for a period of 3 hours at 2 and 3 dpf has no influence on VE-cadherin tension. This may indicate that the mechanical changes controlling EC junctional maturation, if regulated by flow²⁷, become established over a longer developmental timeframe during maturation and that flow plays little role in maintenance of tension through VE-cadherin.”

- abstract: 2nd sentence, should it say: ‘acto-myosin network’?

We have made the necessary textual change.

- page 9: all allele numbers should be included

We have made the necessary textual changes.

Reviewer #2

This manuscript by Legendijk et al. entitled “Live imaging molecular changes in junctional tension upon VE-cadherin in zebrafish” describes the generation and analysis of a VE-cadherin tension sensor zebrafish strain. After validation of protein functionality, the authors demonstrate that VE-cadherin bears mechanical forces in endothelial junctions in an actomyosin-dependent fashion, as described in cell culture experiments before (Conway et al., 2013). In addition, the presented experiments reveal changes in VE-cadherin tension during dorsal aorta maturation that appear to depend on Vegfr2 and calcium signaling. The manuscript is well-written and the experimental observations are interesting. However, experimental controls are missing and it is insufficiently explained how the data have been statistically evaluated. There are also some minor issues that should be corrected. Please see my comments below.

We thank the reviewer these comments and the issues raised below. For a technical paper such as this it is crucial that we carefully control and report the best available controls. We have comprehensively addressed each point below and believe that the manuscript is significantly improved for it.

Major issues:

1. The statistical evaluation of FRET data should be explained in more detail. It is unclear what a single data point in the FRET plots actually stands for (e.g., Fig. 2f, h; Fig.3i, f; Fig.4d, e; Supplementary Fig. 1h; Supplementary Fig. 2d). Is this the average of all junctions from one animal, or one junction? What's the 'n'? How many experimental days are included in these data sets? It will be hard for the reader to evaluate the data without knowing what the data points actually represent and I strongly encourage the authors to be as clear as possible by either writing a separate section on this issue in the methods part, or by including this information directly in the figure legends.

We apologise for the lack of clarity. For energy transfer changes using either ratio-metric FRET or FLIM, each data-point in the graphs corresponds to one junction.

We have now included extensive textual changes throughout the rebuttal and in line with Nature's statistical reporting requirements have made each data-point completely transparent throughout the paper.

2. By generating the VE-cadherin-TL construct, the authors have produced a very useful control to evaluate potential unspecific (i.e. tension independent) effects on FRET. Unfortunately, the control is only used once (Fig.2f) but is missing from all other FRET experiments. How can the authors exclude unspecific effects on FRET in these experiments? I find the lack of a negative control problematic for a number of reasons: First, the authors use ratiometric FRET imaging to determine a FRET index, which is sensitive to the instrumental settings. As expected, the FRET indices vary between experiments and therefore can hardly be compared with each other (e.g. the FRET index of VE-cadherin-TS is about 0.1 in Fig.2f, 0.3 in Fig.2h, 0.45 in Fig.4e; VE-cadherin-TL is about 0.25 in Fig.2f). Second, the authors lack an intermolecular FRET control. Third, Fig. 2e indicates quite a variation of FRET in the VE-cadherin-TL sample, consistent with Fig. 2f which also shows a larger spread in the VE-cadherin-TL data as compared to the VE-cadherin-TS. This may not be a problem per se, but it shows that tension-independent effects are present and should be controlled for. Finally, some FRET differences are rather small (e.g. Fig. 3i, j). Thus, it would significantly strengthen the paper if these experiments were more rigorously controlled. I suggest to include VE-cadherin TL measurements in the experiments described in Fig.2h, Fig.3d, i, j and Fig. 4d, e.

Our responses to the highlighted sections above are separated out and

detailed individually below:

By generating the VE-cadherin-TL construct, the authors have produced a very useful control to evaluate potential unspecific (i.e. tension independent) effects on FRET

We began by revisiting the use of this VE-cadherin-Tailless (VE-cadherin-TL) control. In our previous experiments we had shown that transiently introducing a BAC clone that expresses the TL gives lower tension measurements by FRET compared to transiently introduced TS constructs.

In response to the reviewer's emphasis on this control, we attempted to generate a germline stable negative control transgenic line. We have screened over 60 injected F0 fish and identified 3 founder fish that transmitted (*ve-cad:ve-cadTL*) to their progeny. However we could not detect any VE-cadherin-TL expression in stable transgenic embryos (both using live detection and α -GFP immunofluorescence that binds to Venus).

Next we used DNA injections and upon closer inspection of protein localization following injections of VE-cadherin-TL versus VE-cadherin-TS we found that the TL construct more frequently fails to reach the junctions. We quantified the number of endothelial cells in the dorsal aorta with junctional or non-junctional expression from 3 independent injection rounds. We found that only in 12 out of 32 cells in the dorsal aorta expressing VE-cadherin-TL was protein detected at the cell-cell junctions (37.5%). In 20 out of 32 cells (62.5%) expression was non-junctional and often found as cytoplasmic aggregates likely due to the absence of the β -catenin binding domain (**Supplementary Fig 2g and f'**; **black arrows**). Comparative analysis of endothelial cells in the dorsal aorta expressing VE-cadherin-TS revealed that this full-length construct localized much better with 46 out of 72 cells expressing TS at the junctions (64%) (**Supplementary Fig 2 g and f, black arrowheads**). This demonstrates that there are localization artifacts that upon forced DNA overexpression, especially for the TL construct.

While previous data had measured only junctional localised TL, the more extensive analyses performed in revision suggest that endothelial cells do not recognize TL in the same way as TS and that there are caveats with analyzing such controls.

We still include the original data comparing ratio-metric FRET of junctional TL versus junctional TS expression but this data is now in **Supplementary Figure 2, panels c though to e**. We believe that the potential drawbacks highlighted here are of great importance to the field and we therefore also added discussion of this issue on page 14:

"More difficult to use in vivo was the VE-cadherin Tailless (TL) control construct that

*produced VE-cadherin lacking the β -catenin binding domain and therefore unable to bind to the acto-myosin cytoskeleton. We found that junctional TL did report higher FRET measurements (ie. less tension), however use of the construct was compromised because in many cases it mislocalized in the cell. This is likely due to the β -catenin binding domain being required for efficient VE-cadherin localization, as supported by recent studies in *Drosophila*⁴⁷. Thus, it is possible that such negative control constructs are unreliable and we suggest chemical inhibition of endogenous acto-myosin contractility serves as a more reliable negative control.”*

We agree with the reviewers concerns regarding the need for clear negative controls. Given the issues above, we did not further use the TL as a standard negative control for our major findings. Instead, we utilized the ROCK inhibitor Y-27632 in a series of experiments as a control to probe the role of acto-myosin contractility, which we find to be a more reliable approach (see below points).

How can the authors exclude unspecific effects on FRET in these experiments?

We thank the reviewer for raising this very important point. In response to these comments, we have revisited several controls used and have also now included new controls.

Firstly, ratio-metric FRET imaging is known to be sensitive to fluctuations due to artifacts. Therefore, to investigate if changes we have observed for several key observations using this method can be verified with another method, we performed *in vivo* fluorescent lifetime imaging (FLIM) of *Tg(ve-cad:ve-cadTS)* embryos.

We verified two key points of data using Teal donor lifetime measurements.

1. We performed Teal lifetime imaging on DMSO versus Y-27632 treated *Tg(ve-cad:ve-cadTS)* embryos using the experimental setup as performed in Figure 2f-g We could confirm that tension across VE-cadherin is significantly lower when inhibiting ROCK after a 5-hour treatment with Y-27632 (45 μ M). We have now added these results to **Figure 2 in panels h and i**.
2. Changes in VE-cadherin tension during aorta maturation – Using Teal lifetime measurements we could confirm that tension across VE-cadherin is significantly lower in matured 4 dpf junctions versus more irregular 2 dpf junctions of the dorsal aorta in *Tg(ve-cad:ve-cadTS)* embryos. This data has now been added in **Figure 3, panels f and g**.

We also tried to assess the specificity of Nifedipine calcium inhibition with FLIM but we observed that Nifedipine caused a non-specific Teal lifetime decrease in non-transgenic cells that interferes with any further analysis of the data (**Figure 1 for reviewers**). The same was not observed upon Y-27632 treatment or in the

wild-type 2 dpf versus 4 dpf comparison. Therefore we could not use the FLIM measurements to add to our ratio-metric quantifications of fluorescence (**Figure 1 for reviewers**).

Figure 1: Teal lifetime interference from VE-cadherin-TS negative tissue upon Nifedipine treatment

(a) Heatmap images of Teal lifetime values in VE-cadherin-TS negative tissue surrounding the dorsal aorta of embryos treated with 1% DMSO as a control (n=3, top) and embryos treated with 75µM Nifedipine (n=3, bottom). Colors range from blue (=low lifetime) to red (=high lifetime).

(b) Quantification of Teal lifetime values (ns). Each data point represents a VE-cadherin-TS negative (non-vascular) ROI. For DMSO n=10 non-vascular ROIs from n=8 embryos and for Nifedipine (n=29 non-vascular ROIs from n=8 embryos). Error bars represent mean \pm s.d.; ****p<0.0001, from unpaired two-sided Mann-Whitney-test.

As an independent validation of the existing Nifedipine data, we performed ratio-metric FRET measurements with an additional calcium inhibitor (BAPTA-AM, a cell-permeable calcium quencher). We observed that short term treatments of *Tg(ve-cad:ve-cadTS)* embryos with BAPTA-AM (100µM for 1 hour) similar to Nifedipine (50µM for 30 minutes) induced a significant increase in VE-cadherin tension.

To analyse if changes in tension were acto-myosin dependent we pre-treated embryos with Y-27632 (50µM for 5 hours) followed by a combination of Y-27632 (50µM) and Nifedipine (50µM) for 30 minutes. In these embryos VE-cadherin-TS was under significantly lower tension compared to Nifedipine alone, indicating that changes in tension under low calcium conditions are acto-myosin dependent.

We have now added the analysis of this additional calcium inhibitor and the Y-

27632 rescue experiment in **Figure 5 (panel f)** to complement the previous Nifedipine data.

The authors lack an inter-molecular FRET control

This is a carefully considered technical point from the reviewer and we agree that it would be good to address the influence of intermolecular FRET in VE-cadherin-TS expressing junctions.

To address this question we have sought to determine if increased VE-cadherin-TS expression (measured by Venus fluorescent intensity) correlates with increased FRET/Venus-index in spatially matched pixels since inter-molecular FRET contribution should result in a positive correlation between these quantities
7-11

We first selected regions of interest by drawing lines across single junctions at 2 dpf (**Supplementary Fig. 1e-f' n=2 junctions from separate embryos**) and 4 dpf (**Supplementary Fig. 1g-h' n=2 junctions from separate embryos**). We subsequently plotted the values for Venus intensity and FRET/Venus index within these lines and performed correlation analysis. This analysis of matched pixels within single junctions shows that an increase of VE-cadherin-TS protein does not result in an increase in the FRET/Venus index, both at 2 dpf and 4 dpf.

In agreement with this data we also performed a grouped analysis measuring the average FRET index from a large number of pixels that were grouped based on matching Venus intensity. All pixels were derived from n=52 junctions from n=10 4 dpf embryos (**Supplementary Fig. 1i**).

This independent analysis, like the pixel-matched junctional analysis described previously, revealed that there is no positive correlation between Venus intensity and the Energy transfer and therefore we can exclude presence of inter-molecular FRET that would interfere with the experimental changes in VE-cadherin tension that we report in this study.

We have now added this more comprehensive correlation analysis to **Supplemental Figure 1 (panels e-i)**.

Thus, it would significantly strengthen the paper if these experiments were more rigorously controlled. I suggest to include VE-cadherin TL measurements in the experiments described in Fig.2h, Fig.3d, i, j and Fig. 4d, e.

We agree with the reviewer that addition of a negative control in these

experiments is needed. However, for the reasons outlined above, we do not think that the TL control is the right approach.

We have decided to use the other negative control represented in **Figure 2**, the use of Y-27632 to control our key biological findings that demonstrate the utility of the TS strain. Specifically, we now show with new data that the increased tension seen in the absence of *kdr/kdrl* or after Nifedipine treatment to interfere with calcium signaling, are both dependent on the contractile forces generated by RhoK (as they are rescued with co-treatment with Y-27632). We were unable to use this particular negative control in the case of decreased tension through VE-cadherin over timeframe 2-4 dpf. However in that case we add FLIM measurements at 2 and 4 dpf, which gives additional confidence in the observations.

Overall, performing these experiments *in vivo* has revealed issues with overexpression based negative control constructs and we now discuss this in the discussion section on page 14-15 to aid future efforts in this space.

Minor issues:

1. The references should be checked again. In at least one instance the provided reference does not make much sense in the given context (line 53, ref.7).

We have made the necessary textual changes

2. The numbers of analyzed animals in Fig. 2c, d should be mentioned.

We have made the necessary textual changes

3. In line 472, it should read (j) not (i).

We have made the necessary textual changes

Reviewer #3 This paper from Lagendijk and colleagues describes the introduction of a VE-cadherin tension sensor into the zebrafish vasculature, validation of the construct *in vivo*, and initial assessments of changes to junctional tension during vascular development. The strategy has been previously used in cell culture but this is the first report of a junctional tension sensor in a developing vertebrate. The authors employed an appropriate and well-controlled approach for assessing the FRET-based sensor. The sensor is expressed at endogenous VE-cadherin levels and can rescue VE-cadherin mutants. They show that tension detection requires the sensor's beta-catenin binding site, requires actomyosin activity and is lessened as cell-cell contacts of vasculature smoothen out with development, all properties consistent with published *in vitro* analyses of the sensor. Providing new insights into junctional tension *in vivo*, they report evidence of requirements of Vegfr2 and calcium for

the developmental reduction of tension. The main strength of the paper is the report of a useful tool for studying junctional tension in vivo, a topic of interest in field of tissue morphogenesis. My main concern is the paper's limited conceptual advances, but this concern is typical for a technique-focused paper.

I also have two specific concerns about the conceptual advances.

1. Evidence for the specificity of the applied vegfr2 morpholinos should be provided. Can the effect be rescued with morpholino-resistant mRNA, or can it be seen with mutants?

We agree with the reviewer that it is important to detail the specificity of the morpholino reagents applied to reduce Vegfr2 function.

The morpholinos used here have been previously shown to phenocopy *kdr* and *kdr1*¹² loss of function mutants. *kdr* and *kdr1* double mutants (but not single mutants) result in a complete loss of sprouting phenotype when both *kdr* and *kdr1* are lost¹². This is precisely what we also see with the MOs use in this study, a mutant phenocopy serving as an excellent MO control.

For this study, we aimed to generate a partial loss of Vegfr2 function by knocking down both *kdr* and *kdr1* sub-optimally as complete loss of function results in cardiac defects embryos and a loss of circulation¹². As complete loss of circulation results in aberrant endothelial cell structure, reported here upon *silent heart* knockdown (**Supplementary Fig. 4a**), we needed to analyse embryos with blood flow, therefore mutants would not have been appropriate for these studies. We knocked down both *kdr* and *kdr1* partially and analysed embryos in which arterial sprouting was lost without affecting flow (**Supplementary Fig 3a and Supplementary Video 1**). This is now more clearly described in the methods section on page 20.

To explain our usage of the selected MO reagents we have added the following text on page 9:

*"These MOs phenocopy demonstrated mutant phenotypes for *kdr/kdr1* double mutants³⁰ and we observed single MO phenotypes consistent with the reported, milder, single mutant phenotypes³⁰"*

Furthermore, in response to this reviewer question, we have now included chemical inhibition of signaling downstream of Vegf2 using the very well-validated MEK inhibitor SL327 (4µM)¹³ (see Shin et al 2016. *Development* for full validation of this reagent in zebrafish angiogenesis). We used a dose and treatment staging that allowed for only very mild arterial sprouting defects and did not affect blood circulation. We treated *Tg(ve-cad:ve-cadTS)* embryos from 20 hpf up to 3 dpf and subsequently applied ratio-metric FRET imaging to measure

tension across VE-cadherin. We found that inhibition of MEK by SL327 resulted in a loss of junctional maturation (measured by junctional linearity) and an increase of VE-cadherin tension as observed for *kdr/kdrl* double morphants. Importantly, this new control also suggests that the changes observed are due to signaling downstream of Vegfr2 homologues and not simply due to loss of trunk angiogenesis.

This independent validation of the previous data gives us further confidence in our findings and is now added as **new data to Figure 3 (panels g-i)**.

2. Evidence for the specificity of the applied calcium channel blocker should be provided. Can the effect be seen with a distinct calcium channel blocker?

We agree with the reviewer that an additional calcium inhibitor would strengthen the observations reported here with Nifedipine.

We compared the effect of Nifedipine and BAPTA-AM, a cell-permeable calcium chelator in a short pulse experiment where 3 dpf *Tg(ve-cad:ve-cadTS)* embryos were treated for 1 hour with BAPTA-AM (100 μ M) and imaged immediately after treatment in addition to 1%DMSO and Nifedipine treated embryos (50 μ M for 30 minutes).

BAPTA-AM treatment like Nifedipine induced a significant increase in VE-cadherin tension. These results have now been added as **new data to Figure 4**.

Rebuttal References:

- 1 Bartman, T. *et al.* Early myocardial function affects endocardial cushion development in zebrafish. *PLoS Biol* **2**, E129, doi:10.1371/journal.pbio.0020129 (2004).
- 2 Chen, Q. *et al.* Haemodynamics-driven developmental pruning of brain vasculature in zebrafish. *PLoS Biol* **10**, e1001374, doi:10.1371/journal.pbio.1001374 (2012).
- 3 Steed, E. *et al.* *klf2a* couples mechanotransduction and zebrafish valve morphogenesis through fibronectin synthesis. *Nat Commun* **7**, 11646, doi:10.1038/ncomms11646 (2016).
- 4 Banjo, T. *et al.* Haemodynamically dependent valvulogenesis of zebrafish heart is mediated by flow-dependent expression of miR-21. *Nat Commun* **4**, 1978, doi:10.1038/ncomms2978 (2013).
- 5 Conway, D. E. *et al.* Fluid shear stress on endothelial cells modulates mechanical tension across VE-cadherin and PECAM-1. *Curr Biol* **23**, 1024-1030, doi:10.1016/j.cub.2013.04.049 (2013).

- 6 Sugden, W. W. *et al.* Endoglin controls blood vessel diameter through endothelial cell shape changes in response to haemodynamic cues. *Nat Cell Biol* **19**, 653-665, doi:10.1038/ncb3528 (2017).
- 7 Ferrari, M. L., Gomez, G. A. & Maccioni, H. J. Spatial organization and stoichiometry of N-terminal domain-mediated glycosyltransferase complexes in Golgi membranes determined by fret microscopy. *Neurochem Res* **37**, 1325-1334, doi:10.1007/s11064-012-0741-1 (2012).
- 8 Wallrabe, H. & Periasamy, A. Imaging protein molecules using FRET and FLIM microscopy. *Curr Opin Biotechnol* **16**, 19-27, doi:10.1016/j.copbio.2004.12.002 (2005).
- 9 Kenworthy, A. K., Petranova, N. & Edidin, M. High-resolution FRET microscopy of cholera toxin B-subunit and GPI-anchored proteins in cell plasma membranes. *Mol Biol Cell* **11**, 1645-1655 (2000).
- 10 Kenworthy, A. K. & Edidin, M. Distribution of a glycosylphosphatidylinositol-anchored protein at the apical surface of MDCK cells examined at a resolution of <100 Å using imaging fluorescence resonance energy transfer. *J Cell Biol* **142**, 69-84 (1998).
- 11 Deplazes, E., Jayatilaka, D. & Corry, B. ExiFRET: flexible tool for understanding FRET in complex geometries. *J Biomed Opt* **17**, 011005, doi:10.1117/1.JBO.17.1.011005 (2012).
- 12 Covassin, L. D. *et al.* A genetic screen for vascular mutants in zebrafish reveals dynamic roles for Vegf/Plcg1 signaling during artery development. *Dev Biol* **329**, 212-226, doi:10.1016/j.ydbio.2009.02.031 (2009).
- 13 Shin, M. *et al.* Vegfa signals through ERK to promote angiogenesis, but not artery differentiation. *Development* **143**, 3796-3805, doi:10.1242/dev.137919 (2016).

REVIEWERS' COMMENTS:

Reviewer #1 (Remarks to the Author):

The authors have addressed all of my concerns, and more.

Reviewer #2 (Remarks to the Author):

The authors have performed a range of additional experiments and rewritten parts of the manuscript. As a result, the paper is improved and I recommend publication. Yet, there are a few minor points, which should be clarified. Please see below.

1. The authors state in the introduction: "How broadly applicable such approaches will prove in animal models remains to be determined." It would be helpful to the reader (and fair to colleagues) if it were more clearly stated that a number of TS animal models have been established already (Xenopus, C. elegans, Drosophila), most notably by the Goodman lab.

2. The lifetime data in Fig. 2i suggest very small FRET efficiency differences; the lifetime value for the DMSO sample is at around 2.28 ns, the Y-27632 at 2.25 ns. Assuming the published fluorescence lifetime of 2.65 for teal (in the absence of acceptor), this results in FRET efficiencies of about 14% (DMSO) and 15% (Y-27632). This is really a very small difference and nothing that can be measured easily. Experiments in Fig. 3, 4 and 5 indicate that, in general, the observed FRET changes are quite small for this particular sensor, and I think this should be communicated more clearly. Certainly, it seems important to report the data in Fig. 2i as FRET efficiencies, as this will allow the reader to compare the data with previously published TS FLIM data sets.

3. The lifetime data in Fig. 3f suggest that VE-cadherin tension at 4dpf is reduced below Y-27632 levels. Why is that? Maybe it would be useful to comment on this.

4. The data in Fig. 5f are incorrectly referenced. In line 250 and line 252, the data are referred to as Fig. 4j.

5. I appreciate the discussion on proper controls. It would be helpful to mention the parallel use of TS modules with distinct force sensitivities, as it has been established by the Grashoff lab (Austen et al., 2015) and the Ha lab (Brenner et al, 2016). The application and analysis of distinct TS modules allows not only a more precise quantification of molecular tension, it also serves as a good control to confirm that FRET changes are caused by mechanical forces and not by changes in intermolecular FRET or protein conformation.

Reviewer #3 (Remarks to the Author):

My specific concerns have been addressed effectively. The overall revisions to the paper have improved it both technically and conceptually. I support its publication.

Detailed Response to Reviewer's comments

We thank reviewers 1 and 3 for their very positive response to our manuscript. The remaining issues raised by the second reviewer are addressed below:

Reviewer #2:

The authors have performed a range of additional experiments and rewritten parts of the manuscript. As a result, the paper is improved and I recommend publication. Yet, there are a few minor points, which should be clarified. Please see below.

Questions:

1. The authors state in the introduction: "How broadly applicable such approaches will prove in animal models remains to be determined." It would be helpful to the reader (and fair to colleagues) if it were more clearly stated that a number of TS animal models have been established already (Xenopus, *C. elegans*, *Drosophila*), most notably by the Goodman lab.

We agree that it is important to clarify in more detail the current applications of tension sensor models in various organisms and we thank the reviewer for pointing out we mistakenly overlooked referencing the elegant application of TS technology in *C. elegans* by the Goodman lab.

We have amended the introduction on page 3-4 to include these applications in more detail:

*"Tension Sensor (TS) technology has introduced an optical imaging-based read-out for tension across individual molecules in cultured cells^{5,8-10}. Such imaging-based quantification of intra-molecular tension through key adhesive molecules holds great promise for direct observation of forces that control morphogenesis. Recent applications of functionally validated TS-proteins for β -spectrin in *C. elegans*¹¹ and E-cadherin in *Drosophila*¹² have confirmed the potential of measuring intra-molecular tension changes in invertebrate models."*

And in the discussion on page 12-13:

"To our knowledge this strain represents the first functionally tested, tissue restricted TS system demonstrated to measure tensile changes in a vertebrate. Hence, we expand on previous overexpression of an α -actinin-TS in frogs⁴⁷ and highlight the potential for translation of Tension biosensors to other vertebrates, tissues, cellular structures and disease models."

2. The lifetime data in Fig. 2i suggest very small FRET efficiency differences; the lifetime value for the DMSO sample is at around 2.28 ns, the Y-27632 at 2.25 ns. Assuming the published fluorescence lifetime of 2.65 for teal (in the absence of

acceptor), this results in FRET efficiencies of about 14% (DMSO) and 15% (Y-27632). This is really a very small difference and nothing that can be measured easily. Experiments in Fig. 3, 4 and 5 indicate that, in general, the observed FRET changes are quite small for this particular sensor, and I think this should be communicated more clearly. Certainly, it seems important to report the data in Fig. 2i as FRET efficiencies, as this will allow the reader to compare the data with previously published TS FLIM data sets.

We thank the reviewer for making these points, which are completely correct. The scale of the differences in FRET efficiency measured by FLIM are small compared with some published datasets¹⁻⁴. Specifically, our Y-27632 treatment led to increases of 1%, our measurements comparing 2 dpf and 4 dpf junctions revealed a change of 4%. While these changes are subtle compared, the later is in line with the previously published lifetime measurements using the VE-cadherin sensor *in vitro*⁵.

We have amended the text in the results section to reflect this and to highlight that the differences measured, while completely reliable to draw biological conclusions, reflect small changes in overall FRET efficiency:

Page 9:

"Of note, the differences in FRET efficiency measured by FLIM were small. The efficiency change for Y-27632 compared with DMSO was 1% and comparing 2 dpf to 4 dpf junctions was 4% (see Methods). Interestingly, cultured ECs align with the direction of flow when under shear²⁸ and this was shown to coincide with decreased tension through VE-cadherin⁵, measured by FLIM as a 4% increase in FRET efficiency using a VE-cadherin-TS. This data is in line with the efficiency changes measured here in aligned, linear junctions at 4 dpf compared to immature junctions at 2 dpf. In addition, previously published comparison of a negative control construct with the VE-cadherin-TS revealed only a 7% change in FRET efficiency⁵ measured by FLIM. Together, these observations suggest that the dynamic range for this VE-cadherin-TS is limited. Nevertheless, these data combined demonstrate that we can measure meaningful changes in tension through VE-cadherin in vivo."

3. The lifetime data in Fig. 3f suggest that VE-cadherin tension at 4dpf is reduced below Y-27632 levels. Why is that? Maybe it would be useful to comment on this.

We have added the following sentence to the results on page 7-8, which we feel directly addresses this question:

Treatment with Y-27632 was carefully titrated to avoid induction of developmental defects. Therefore, while it acts to reduce acto-myosin contractility it may not be a fully inhibitory dosage."

4. The data in Fig. 5f are incorrectly referenced. In line 250 and line 252, the data are referred to as Fig. 4j.

We have made the necessary textual change.

5. I appreciate the discussion on proper controls. It would be helpful to mention the parallel use of TS modules with distinct force sensitivities, as it has been established by the Grashoff lab (Austen et al., 2015) and the Ha lab (Brenner et al, 2016). The application and analysis of distinct TS modules allows not only a more precise quantification of molecular tension, it also serves as a good control to confirm that FRET changes are caused by mechanical forces and not by changes in intermolecular FRET or protein conformation.

We thank the reviewer for this important suggestion. We agree that the application of multiple TS modules with different mechanical sensitivities in the future will improve the quantitative nature of the approach.

We have expanded our current discussion on TS modules on page 12-13:

“We expect that ongoing optimization of TS modules with different force sensitivities^{11,13,14} as well as improved sensitivity in microscopy in the future will allow us to improve this and to quantify intra-molecular tension and contractile force with increasing sensitivity.”

Rebuttal References:

- 1 Krieg, M., Dunn, A. R. & Goodman, M. B. Mechanical control of the sense of touch by beta-spectrin. *Nat Cell Biol* **16**, 224-233, doi:10.1038/ncb2915 (2014).
- 2 Grashoff, C. *et al.* Measuring mechanical tension across vinculin reveals regulation of focal adhesion dynamics. *Nature* **466**, 263-266, doi:10.1038/nature09198 (2010).
- 3 Brenner, M. D. *et al.* Spider Silk Peptide Is a Compact, Linear Nanospring Ideal for Intracellular Tension Sensing. *Nano Lett* **16**, 2096-2102, doi:10.1021/acs.nanolett.6b00305 (2016).
- 4 Austen, K. *et al.* Extracellular rigidity sensing by talin isoform-specific mechanical linkages. *Nat Cell Biol* **17**, 1597-1606, doi:10.1038/ncb3268 (2015).
- 5 Conway, D. E. *et al.* Fluid shear stress on endothelial cells modulates mechanical tension across VE-cadherin and PECAM-1. *Curr Biol* **23**, 1024-1030, doi:10.1016/j.cub.2013.04.049 (2013).